# GAIA: Delving into Gradient-based Attribution Abnormality for Out-of-distribution Detection

**Jinggang Chen**[†*]**, Junjie Li**[†*]**, Xiaoyang Qu**[‡§]**, Jianzong Wang**[‡§]**, Jiguang Wan**[†]**, Jing Xiao**[‡]

[†] Huazhong University of Science and Technology, China
[‡] Ping An Technology (Shenzhen) Co., Ltd.
{chen.jinggang98, 2216217669ljj, quxiaoy}@gmail.com, jzwang@188.com,
jgwan@hust.edu.cn, xiaojing661@pingan.com.cn

## Abstract

Detecting out-of-distribution (OOD) examples is crucial to guarantee the reliability and safety of deep neural networks in real-world settings. In this paper, we offer an innovative perspective on quantifying the disparities between in-distribution (ID) and OOD data—analyzing the uncertainty that arises when models attempt to explain their predictive decisions. This perspective is motivated by our observation that gradient-based attribution methods encounter challenges in assigning feature importance to OOD data, thereby yielding divergent explanation patterns. Consequently, we investigate how attribution gradients lead to uncertain explanation outcomes and introduce two forms of abnormalities for OOD detection: the zero-deflation abnormality and the channel-wise average abnormality. We then propose **GAIA**, a simple and effective approach that incorporates **G**radient **A**bnormality **I**nspection and **A**ggregation. The effectiveness of GAIA is validated on both commonly utilized (CIFAR) and large-scale (ImageNet-1K) benchmarks. Specifically, GAIA reduces the average FPR95 by 23.10% on CIFAR10 and by 45.41% on CIFAR100 compared to advanced post-hoc methods.

## 1 Introduction

Deep neural networks have been extensively applied across various domains, demonstrating remarkable performance. However, when they are deployed in real-world scenarios, particularly in contexts that require high levels of security [1–3], an urgent challenge arises. Namely, these models must be able to ensure the reliability of their outcomes, even in the face of out-of-distribution (OOD) inputs from the open world that differ from in-distribution (ID) training data and thus surpass their cognitive capabilities. That underscores the importance of OOD detection, which involves estimating uncertainty from the model to identify the "unknown" samples, serving as an alert mechanism before making predictive decisions.

Recently, a rich line of literature has emerged to address the challenge of OOD detection [4–13]. Indeed, the majority of previous approaches focus on defining more suitable measures of OOD uncertainty by using model outputs [4, 5, 7–9] or feature representations [11, 13–15]. Despite the above mainstream approaches, estimating uncertainty from gradients is readily implemented with a fixed model and has received increasing research attention lately. Prior gradient-based OOD detection methods [10, 16, 17] have primarily emphasized utilizing parameter gradients as the measurement, while giving limited attention to the in-depth exploration of gradients related to the inputs (*i.e.*, attribution gradients [18]).

---

[*]Equal Contribution.
[§]Corresponding Author.

37th Conference on Neural Information Processing Systems (NeurIPS 2023).

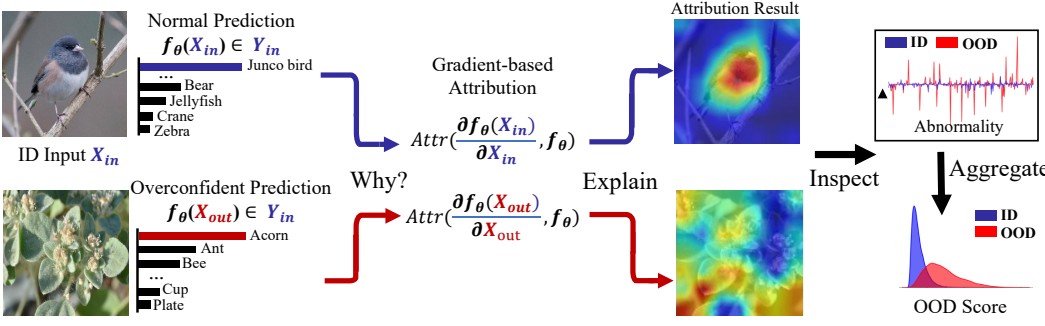

Figure 1: Motivation of our work. Gradient-based attribution algorithms use attribution gradients to explain where models look for predicting final outputs. An intriguing question is: when encountering OOD sample $X_{out}$ whose label falls outside the in-distribution label space $Y_{in}$, how does the model interpret its overconfident prediction? In order to unearth uncertainty from the explanatory result, we conduct our research by inspecting the abnormalities in attribution gradients and then aggregate them for OOD detection.

In this paper, we put our eye on a novel and insightful perspective — let models explain the uncertainty themselves with attribution approaches. Gradient-based attribution algorithms [19–21] are ubiquitous for the visual explanation of why the model makes such a decision to the predicted class. An intuition comes up that well-trained networks can clearly attribute the region of target ID objects, but what if they face OOD samples that are totally unknown to them? As shown in Fig. 1, we observe through the utilization of attribution gradients that the pre-trained model is capable of generating reasonable visual interpretation for the ID input $X_{in}$ from ImageNet [22]. However, when attempting to interpret an OOD image $X_{out}$ from iNaturalist [23] with a label that does not belong to $Y_{in}$, it confuses the model, leading to a meaningless attribution result.

Following the observation, we delve into investigating the gradient-based attribution abnormality when inferring OOD examples. Our further study finds that this phenomenon can be caused by the attribution gradient, which is constructed by taking the value of the partial derivative of the target output $S_c(\cdot)$ *w.r.t.* one unit $z_i$ of the input variables $z$ (*i.e.*, $\frac{\partial S_c(z)}{\partial z_i}$). To enlarge the discrepancy between ID and OOD without prior knowledge from training data, we introduce the channel-wise average abnormality and the zero-deflation abnormality as two measurements for detecting distributional shifts. Then, we propose our detection framework **GAIA** with **G**radient **A**bnormality **I**nspection and **A**ggregation and conduct comprehensive experiments on both CIFAR benchmarks and large-scale ImageNet-1K benchmark to validate the effectiveness of our proposed method. Code is available at https://github.com/JGEthanChen/GAIA-OOD.

Our key results and contributions are summarized as follows:

- We provide insights into the attribution abnormality for OOD detection. Our intuition is that unreliability from visual explanations can be a direct alarm to distinguish OOD examples. Hence, we delve further into the underlying causality of the abnormality. Then, we provide a theoretical explanation for the causes of attribution abnormality.

- We propose a simple yet effective post-hoc detection framework via **G**radient **A**bnormality **I**nspection and **A**ggregation (**GAIA**), which consists of two independent measurements: the Channel-wise Average abnormality (**GAIA-A**) and the Zero-deflation abnormality (**GAIA-Z**). Both of them are lightweight and plug-and-play—hyperparameter-free, training-free, with no ID data and outliers required for estimation.

- Thorough experiments demonstrate that GAIA surpasses most advanced post-hoc methods on both commonly utilized (CIFAR) and large-scale (ImageNet-1K) benchmarks. GAIA-Z exhibits superior performance on CIFAR benchmarks, reducing the average FPR95 by 23.10% on CIFAR10 and by 45.41% on CIFAR100. GAIA-A performs well on the ImageNet-1K benchmark and reduces by 17.28% compared to the advanced gradient-based detection method GradNorm.

## 2   Preliminaries

We consider the general setting of a supervised machine learning problem, where $\mathcal{X}$ denotes the input space and $\mathcal{Y}_{\text{in}} = \{1, 2, ..., C\}$ denotes the ID label space. Especially, we denote the output score *w.r.t.* class $c$ before softmax layer as $S_c(\cdot)$.

**Out-of-distribution detection.** The goal of out-of-distribution (OOD) detection is to distinguish the sample $x_{\text{out}}$ that exhibits substantial deviation from the distribution $\mathcal{X}$. In literature, OOD data originates from an unknown distribution $\mathcal{X}_{\text{out}}$. And the label space of the OOD samples has no intersection with $\mathcal{Y}_{\text{in}}$. This problem can be formulated as a binary classification task using a score function $\Delta(x)$. More specifically, when provided with an input sample $x$, the level-set estimation can be expressed as follows:

$$\mathcal{G}(x) = \begin{cases} \text{out}, & \text{if} \quad \Delta(x) > \gamma \\ \text{in}, & \text{if} \quad \Delta(x) \le \gamma \end{cases} \tag{1}$$

In our work, lower scores correspond to a higher likelihood of classifying the sample $x$ as in-distribution (ID), and $\gamma$ denotes a threshold for separating the ID and OOD data.

**Gradients from attribution algorithms.** The attribution gradient is first introduced by sensitivity analysis (SA) [18] and widely utilized in visual explainability techniques [19–21, 24, 25]. It refers to the sensitivity of a particular input variable (input or feature unit) *w.r.t.* $c$-class predictive output $S_c(\cdot)$. Denotes $k$-th channel feature map at layer $l$ as $\boldsymbol{A}^{kl} \in \mathbb{R}^{W \times H}$. The attribution gradient of one feature unit $A_{ij}^{kl}$ is computed by:

$$\text{Grad}_{ij} = \frac{\partial S_c(\boldsymbol{A}^{kl})}{\partial A_{ij}^{kl}} \tag{2}$$

It is unrelated to the gradients commonly associated with the typical understanding of network optimization (*i.e.*, gradients of the parameters). In most attribution algorithms, the attribution gradient is used for quantifying the contribution of each feature unit to the model's prediction.

## 3   Investigating Attribution Abnormality for Out-of-distribution Detection

In this section, we aim to investigate how attribution gradients can lead to abnormality when explaining OOD examples. We also attempt to provide a unified theoretical analysis.

**Channel-wise average abnormality.** We first focus on the abnormality in the Gradient-based Class Activation Mapping (GradCAM) algorithm [19], which is one of the most widely applied attribution strategies. Its paradigm is to channel-wise sum up feature maps for a saliency map $\boldsymbol{M} \in \mathbb{R}^{W \times H}$. Here we denote feature maps $\boldsymbol{A} \in \mathbb{R}^{K \times W \times H}$ with $K$-channels in the convolutional layer as the input variables. The attribution $\alpha_{ij}$ of each unit $M_{ij}$ can be formulated as follows:

$$\alpha_{ij} = \text{ReLU}(\sum_{k=1}^{K} w_k A_{ij}^k), \quad \text{where} \quad w_k = \frac{\partial g(\boldsymbol{A})}{\partial A_{ij}^k} = \frac{1}{W \times H} \sum_{i=1}^{W} \sum_{j=1}^{H} \frac{\partial S_c(\boldsymbol{A})}{\partial A_{ij}^k} \tag{3}$$

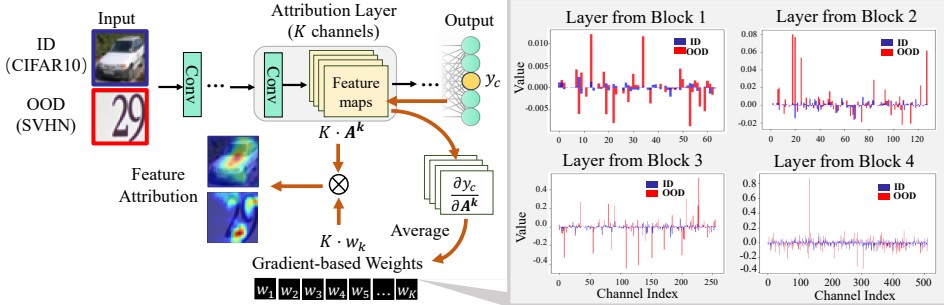

Figure 2: Demonstration of the attribution abnormality from gradient-based weights. The toy experiment is conducted on ResNet34 with four blocks trained on CIFAR10. We select four attribution layers from different blocks and calculate the average attribution gradients for each channel.

where $w_k$ is the channel-wise weight that re-weights feature maps in different channels, and $g(\boldsymbol{A})$ denotes the explanatory function of the DNN output from $\boldsymbol{A}$. Detailed elaboration is provided in *Appendix C*. Taking different layers as the attribution targets, we visualize the distribution of channel-wise average attribution gradients in Fig. 2. It can be observed that the discrepancy of the weights $w_k$ is distinguishable in that OOD samples tend to produce more noisy and abnormal outliers compared to ID samples. Additionally, as the layers increase in depth, the magnitude of the average gradients also increases.

**Zero-deflation abnormality.** Then, we closely examine the abnormality that may arise in attribution gradients themselves due to distributional shifts. Fig. 3(a) shows attribution gradients on feature maps across all channels at a specific layer. We observe that the quantity of zero partial derivation $\frac{\partial S_c(\boldsymbol{A})}{\partial A_{ij}^k}$ in OOD is extremely less than ID, leading to a high occurrence of dense gradient matrices. As shown in Fig. 3(b), this phenomenon is more pronounced in deeper layers, indicating an abnormal behavior.

## 3.1 Theoretical Explanation for Attribution Abnormality

We consider a unified explanation for attribution algorithms with Taylor expansion. As proved in [26], attribution algorithms are mathematically equivalent to the perspective that the network $c$-class output $S_c(\boldsymbol{z})$ is explained as a Taylor expansion model. For variables $\boldsymbol{z} = [z_1, ..., z_n]$ (*e.g.*, feature units to be attributed or inputs), here we perform $P$-order expansion of zero baseline output $S_c(\boldsymbol{0})$ at $\boldsymbol{z}$:

$$S_c(\boldsymbol{0}) = S_c(\boldsymbol{z}) + \sum_{p=1}^{P}\sum_{i=1}^{n}\frac{1}{p!}\frac{\partial^p S_c(\boldsymbol{z})}{\partial(z_i)^p}(z_i)^p + \frac{1}{2!}\frac{\partial^2 S_c(\boldsymbol{z})}{\partial z_1 \partial z_2}z_1 z_2 + ... + R_P(\boldsymbol{z}) \quad (4)$$

where $R_P(\boldsymbol{z})$ denotes the remainder term for the $P$-order expansion. In our paper, we consider the feature values to be all zeros as the zero baseline, which is commonly adopted for analyzing gradient-based attribution algorithms. Then all terms can be represented by vector $\boldsymbol{\kappa} = [\kappa_1, ..., \kappa_n] \in \mathbb{N}^n$, where $\kappa_i \in \mathbb{N}$ reflects the integral degree of the input variable $z_i$ (*e.g.*, $\kappa_i = 1$ indicates the corresponding item only contains first-order partial derivative *w.r.t.* $z_i$). Thus we can represent the $c$-label output change caused by variables $\boldsymbol{z}$ as:

$$|S_c(\boldsymbol{z}) - S_c(\boldsymbol{0})| = |\sum_{p=1}^{P}\sum_{\boldsymbol{\kappa}\in\Omega, |\boldsymbol{\kappa}|=p}\mathcal{C}(\boldsymbol{\kappa})\cdot\frac{\partial^{\kappa_1+\cdots+\kappa_n}S_c(\boldsymbol{z})}{\partial^{\kappa_1}z_1\cdots\partial^{\kappa_n}z_n}(z_1)^{\kappa_1}\cdots(z_n)^{\kappa_n} + R_P(\boldsymbol{z})| \quad (5)$$

where $\mathcal{C}(\boldsymbol{\kappa})$ is a non-negative constant related to vector $\boldsymbol{\kappa}$. The expansion formula reflects the contribution of each variable $z_i$ to the $c$-label output change. Thus, we can attribute importance $\alpha_i$ to $z_i$ based on how much it contributes to such a change. Furthermore, the effect of $z_i$ to $S_c(\boldsymbol{z})$ can be decomposed into Taylor independent effect term $\phi_i(\boldsymbol{\kappa})$ and Taylor interaction effect term $\psi_i(\boldsymbol{\kappa})$. For independent term $\phi_i(\boldsymbol{\kappa})$, only $z_i$ is contained, where $\boldsymbol{\kappa} = [0, \cdots, \kappa_i, \cdots, 0] \in \Omega_\phi$ and $\kappa_i > 0$. And

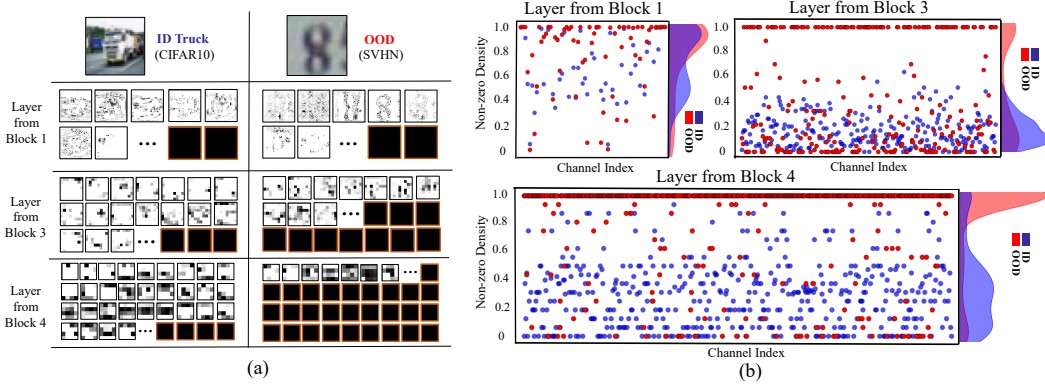

Figure 3: **Left (a):** Visualization of attribution gradients on feature maps. **Right (b):** Proportion of non-zero values across different channels. Each data point represents one single channel.

the overall effect of term $\psi_i(\boldsymbol{\kappa})$ is caused by the interactions between $z_i$ and other variables, where $\boldsymbol{\kappa} \in \Omega_\psi$ has at least two non-negative values and $\kappa_i > 0$. Attribution methods are formulated as:

$$\alpha_i = \sum_{p=1}^{P} \sum_{\boldsymbol{\kappa} \in \Omega_\phi, |\boldsymbol{\kappa}|=p} \omega_{i,\boldsymbol{\kappa}} \phi_i(\boldsymbol{\kappa}) + \sum_{p=1}^{P} \sum_{\boldsymbol{\kappa} \in \Omega_\psi, |\boldsymbol{\kappa}|=p} \omega_{i,\boldsymbol{\kappa}} \psi_i(\boldsymbol{\kappa}) \tag{6}$$

where $\omega_{i,\boldsymbol{\kappa}}$ denotes the ratio of a specific term (either the independent term or the interaction term) allocated to $\alpha_i$.

**Attribution abnormality in zero importance.** A reliable attribution result requires accurate identification of the features that are useful for the output. Here, we consider the Null-player axiom [27] (see *Appendix E*), which states that in the reliable attribution, a feature should be considered as having zero importance when it makes no contribution to the model's output. In other words, *if a feature does not contribute to the model's prediction, it should be considered as having zero importance.*

**Proposition 1.** *Given input variables $\boldsymbol{z}$, for one variable $z_i \in \boldsymbol{z}$ to be attributed, if $\frac{\partial S_c(\boldsymbol{z})}{\partial z_i}$ is zero throughout the analysis, then $\alpha_i = 0$ always holds.*

Given variables $\boldsymbol{z}$ in one analysis, it is assumed that the partial derivative function *w.r.t.* $z_i$ is a constant zero. As shown in Eq. 7, all independent and interaction terms related to $z_i$ are zero. Thus, $z_i$ is of zero importance to the prediction. This is, zero attribution gradient values will directly impact the final attribution result.

$$\frac{\partial S_c(\boldsymbol{z})}{\partial z_i} = 0 \Rightarrow \frac{\partial^{\kappa_1 + \cdots + \kappa_n} S_c(\boldsymbol{z})}{\partial^{\kappa_1} z_1 \cdots \partial^{\kappa_n} z_n} = 0, \kappa_i > 0 \Rightarrow \phi_i(\boldsymbol{\kappa}) = \psi_i(\boldsymbol{\kappa}) = 0 \Rightarrow \alpha_i = 0 \tag{7}$$

This provides us with an explanatory perspective for our observation — visual explanation for OOD data tends to be messy and unreliable due to the model's uncertainty about the unknown distribution, resulting in an abundance of intricate non-zero importance attributions.

**Attribution abnormality in gradient-based weights.** Following Eq. 6, GradCAM in Eq. 3 can be reformulated in form that includes only the first-order Taylor independent terms (see *Appendix D* for the proof), where $\boldsymbol{\kappa} = [0, ..., \kappa_i = 1, ...0]$ is a one-hot vector, and $\kappa_j = 0$ if $j \neq i$. This simplifies our analysis of the abnormality in weights, focusing solely on the correlation between first-order partial derivatives and the attribution result to reflect the uncertainty on each independent feature.

## 4 GAIA: A Simple and Effective Framework for Abnormality Aggregation

We propose our GAIA framework, which aggregates the channel-wise average abnormality (GAIA-A) or the zero-deflation abnormality (GAIA-Z) for out-of-distribution detection.

**Abnormality aggregation from label space.** General attribution algorithms focus on the final predictive output $S_c(\boldsymbol{A})$, where $c = \mathrm{argmax}_{c_i \in C} S_{c_i}(\boldsymbol{A})$. This is adequate for the zero-deflation abnormality as we aim to ascertain the model's confidence in interpreting its own classification result. While for the channel-wise average abnormality, our aspiration is to gather abnormalities from a broader label space. Hence, all outputs in the ID label space are informative for collecting the model's tendency towards identifying samples as ID categories. For GAIA-A, we fuse all the outputs with $\log(\mathrm{softmax}(\cdot))$:

$$\frac{\partial \mathcal{S}(\boldsymbol{A})}{\partial A_{ij}^k} = \frac{\partial \sum_{c \in C} \log \mathrm{softmax}(S_c(\boldsymbol{A}))}{\partial A_{ij}^k} \tag{8}$$

This strategy first accumulates the model's outputs and simultaneously performs backpropagation *w.r.t.* the features $A_{ij}^k$. It is more efficient compared to individually backpropagating through each category and then accumulating them, which is impractical in scenarios with large label space (*e.g.*, 1000 categories in ImageNet). Furthermore, we find that GAIA-A can be enhanced with a two-stage fusion strategy. Let us denote the neural network prediction function based on input feature variables $\boldsymbol{A} \in \mathbb{R}^{K \times W \times H}$ by $S_c(\boldsymbol{A}) = \Psi(\boldsymbol{A}_{\mathrm{last}}, \Theta_\Psi)$, where $\Psi(\cdot)$ represents the classification function and $\boldsymbol{A}_{\mathrm{last}} \in \mathbb{R}^{1 \times W_{\mathrm{last}} \times H_{\mathrm{last}}}$ is the feature map at the last layer. Then the network feature extraction function is defined as $\Phi(\cdot)$ and $\boldsymbol{A}_{\mathrm{last}} = \Phi(\boldsymbol{A}, \Theta_\Phi)$. In our methods, we consider the gradient matrix on the $\boldsymbol{A}_{\mathrm{last}}$ and the inner feature map $\boldsymbol{A}^k \in \mathbb{R}^{1 \times W \times H}$ ($k$-th channel from $\boldsymbol{A}$) separately, with the former

regarded as the output component $\nabla \boldsymbol{A}_{\text{last}}$ and the latter as the inner component $\nabla \boldsymbol{A}^k$:

$$\nabla \boldsymbol{A}^k = \frac{\partial \Phi(\boldsymbol{A}, \Theta_\Phi)}{\partial \boldsymbol{A}^k}$$

$$\nabla \boldsymbol{A}_{\text{last}} = \frac{\partial \mathcal{S}(\boldsymbol{A}_{\text{last}})}{\partial \boldsymbol{A}_{\text{last}}} = \frac{\partial \sum_{c \in C} \log \text{softmax}(S_c(\boldsymbol{A}_{\text{last}}))}{\partial \boldsymbol{A}_{\text{last}}} \tag{9}$$

**Abnormality aggregation from input space.** We start by defining the anormalies expectation on $k$-th channel feature map at $l$ layer as $\boldsymbol{A}^{kl} \in \mathbb{R}^{1 \times W \times H}$. The zero-deflation abnormality can be described as the non-zero density of $\boldsymbol{A}^{kl}$:

$$\mathbb{E}[\epsilon | \boldsymbol{A}^{kl}] = \frac{1}{W \times H} \mid \{A_{ij}^{kl} \quad \mid \quad \frac{\partial S_c(\boldsymbol{A}^{kl})}{\partial A_{ij}^{kl}} \neq 0\} \mid \tag{10}$$

For the channel-wise average abnormality, we observed that average gradients on $\boldsymbol{A}_{\text{last}}$ from the output component and the average attribution gradients obtained from the inner component exhibit opposite behaviors in terms of ID and OOD data *(We discuss its effectiveness in Section 5.3 and provide theoretical analysis in Appendix F)*. Consequently, we use division to get the expectation of the final fusion channel-wise average abnormality abnormality:

$$\mathbb{E}[\epsilon | \boldsymbol{A}^{kl}] = \frac{\mathbb{E}_{\text{inner}}[\epsilon | \boldsymbol{A}^{kl}]}{\sqrt{\mathbb{E}_{\text{output}}[\epsilon | \boldsymbol{A}_{\text{last}}]}} = \frac{\mid \frac{1}{W \times H} \sum_{g^{kl} \in \nabla \boldsymbol{A}^{kl}} g^{kl} \mid}{\mid \frac{1}{W_{\text{last}} \times H_{\text{last}}} \sum_{g_{\text{last}} \in \nabla \boldsymbol{A}_{\text{last}}} g_{\text{last}} \mid^{\frac{1}{2}}} \tag{11}$$

Consider networks have $L$ layers to be utilized, and each layer has $K_l$ channels. Our framework accumulates them into an abnormality matrix $\boldsymbol{\Lambda} \in \mathbb{R}^{L \times K_m}$, where $K_m = \max\{K_i | 1 \leq i \leq L\}$ and $\Lambda_{ij} = 0$ if $j > K_i$. Then, we use the Frobenius norm as a non-parameter measuring score to represent the global abnormality. For instance, assuming $K_m = K_L$, $\|\boldsymbol{\Lambda}\|_F$ is calculated as:

$$\|\boldsymbol{\Lambda}\|_F = \left\| \begin{matrix} \mathbb{E}[\epsilon | \boldsymbol{A}^{1,1}] & \cdots & \mathbb{E}[\epsilon | \boldsymbol{A}^{1,K_1}] & 0 & \cdots & 0 \\ \vdots & & \vdots & & & \vdots \\ \mathbb{E}[\epsilon | \boldsymbol{A}^{L,1}] & \cdots & \mathbb{E}[\epsilon | \boldsymbol{A}^{L,K_1}] & \cdots\cdots\cdots\cdots & \mathbb{E}[\epsilon | \boldsymbol{A}^{L,K_m}] \end{matrix} \right\|_F = \sqrt{\sum_i^L \sum_j^{K_m} (\mathbb{E}[\epsilon | \boldsymbol{A}^{i,j}])^2} \tag{12}$$

The overall process are formulized in Algorithm 1.

---

**Algorithm 1:** GAIA

---

**Input:** Test sample $x$; Fixed model $f_\theta$.
**Output:** OOD score $\Delta(x)$.

Compute label output set $\{S_c(x) | c \in C\}$ by $f_\theta(x)$;

Backpropagate attribution gradients by $\frac{\partial S_c(\boldsymbol{A}^{kl})}{\partial A_{ij}^{kl}}$ (GAIA-Z) or Eq. 9 (GAIA-A);

Calculate $\mathbb{E}[\epsilon | \boldsymbol{A}^{kl}]$ by Eq. 10 (GAIA-Z) or Eq. 11 (GAIA-A);
Calculate global abnormality $\|\boldsymbol{\Lambda}\|_F$ by Eq. 12;
**return** $\|\boldsymbol{\Lambda}\|_F$ as OOD score $\Delta(x)$.

---

## 5 Experiments

In this section, we describe our experimental setup in Section 5.1. Then, we demonstrate the effectiveness of our method on the large-scale ImageNet-1K benchmark [28] and the CIFAR benchmarks [7] in Section 5.2. We also conduct ablation studies in Section 5.3.

### 5.1 Setup

**Benchmarks.** In accordance with [10, 11, 15, 28], we employ the large-scale ImageNet-1K benchmark [28], which offers a more realistic and challenging environment due to its use of high-resolution images and an large label space that encompasses 1,000 distinct categories. Four OOD datasets in this

| ID Datasets | Methods | SVHN | | TinyImageNet | | LSUN | | Places | | Textures | | Average | |
|---|---|---|---|---|---|---|---|---|---|---|---|---|---|
| | | FPR95 (↓) | AUROC (↑) | FPR95 (↓) | AUROC (↑) | FPR95 (↓) | AUROC (↑) | FPR95 (↓) | AUROC (↑) | FPR95 (↓) | AUROC (↑) | FPR95 (↓) | AUROC (↑) |
| CIFAR10 (ResNet34) | MSP [7] | 61.03 | 89.01 | 53.11 | 85.79 | 46.79 | 90.63 | 43.71 | 91.88 | 48.28 | 90.08 | 50.58 | 89.48 |
| | ODIN [9] | 50.74 | 92.09 | 39.82 | 92.62 | 33.34 | 94.17 | 36.53 | 93.18 | 45.00 | 91.11 | 41.09 | 92.63 |
| | Energy [5] | 42.87 | 91.20 | 37.76 | 92.98 | 34.25 | 93.85 | 38.34 | 92.44 | 45.73 | 90.26 | 39.79 | 92.15 |
| | Mahalanobis [14] | 22.19 | 93.36 | 29.35 | 90.16 | 25.31 | 91.89 | 28.61 | 91.26 | 39.34 | 87.02 | 28.96 | 90.74 |
| | ReAct [11] | 24.60 | 92.39 | 33.68 | 89.71 | 19.15 | 93.78 | 23.69 | 92.78 | 32.61 | 89.27 | 26.75 | 91.59 |
| | GradNorm [10] | 62.47 | 76.08 | 73.00 | 65.21 | 59.38 | 72.97 | 58.93 | 75.36 | 67.77 | 67.41 | 64.31 | 71.41 |
| | KNN [29] | 32.03 | 95.28 | 29.56 | 95.44 | 27.42 | 95.92 | 41.77 | 93.26 | 35.41 | 94.87 | 33.24 | 94.95 |
| | Rankfeat [15] | 84.58 | 72.99 | 50.20 | 89.84 | 41.63 | 91.97 | 67.79 | 82.64 | 68.12 | 80.67 | 62.46 | 83.62 |
| | ASH-P@70 [30] | 23.11 | 95.53 | 29.78 | 93.71 | 22.72 | 95.33 | 25.27 | 94.35 | 30.92 | 93.08 | 26.36 | 94.40 |
| | **GAIA-Z (Ours)** | **2.47** | **99.49** | **6.26** | **98.63** | **2.48** | **99.43** | **2.27** | **99.50** | **2.84** | **99.36** | **3.26** | **99.28** |
| | **GAIA-A (Ours)** | _14.44_ | _97.12_ | _16.45_ | _97.07_ | _9.10_ | _98.10_ | _11.06_ | _97.82_ | _12.62_ | _97.54_ | _12.73_ | _97.53_ |
| CIFAR10 (WRN40) | MSP [7] | 40.51 | 92.70 | 50.05 | 86.99 | 38.90 | 91.34 | 45.41 | 89.58 | 56.42 | 84.57 | 46.26 | 89.04 |
| | ODIN [9] | 16.11 | 96.91 | 44.18 | 89.66 | 33.37 | 93.45 | 40.30 | 91.31 | 51.51 | 87.71 | 37.09 | 91.81 |
| | Energy [5] | 19.94 | 95.80 | 41.70 | 90.04 | 37.95 | 91.44 | 44.88 | 89.67 | 55.89 | 84.58 | 40.07 | 90.31 |
| | Mahalanobis [14] | 21.63 | 94.99 | 42.86 | 89.77 | 46.87 | 86.58 | 45.39 | 89.47 | 48.06 | 88.65 | 40.96 | 89.89 |
| | ReAct [11] | 20.05 | 95.87 | 41.32 | 90.29 | 37.81 | 91.57 | 44.28 | 89.77 | 54.88 | 85.54 | 39.67 | 90.61 |
| | GradNorm [10] | 49.60 | 80.45 | 82.23 | 59.60 | 78.17 | 63.55 | 81.70 | 59.72 | 82.93 | 58.05 | 74.93 | 64.27 |
| | KNN [29] | 27.52 | 95.55 | 38.14 | 93.44 | 38.95 | 94.30 | 45.79 | 90.65 | 50.37 | 90.77 | 40.16 | 92.94 |
| | Rankfeat [15] | 60.02 | 72.03 | 72.33 | 63.24 | 52.17 | 83.24 | 78.43 | 61.27 | 86.22 | 51.97 | 69.83 | 66.35 |
| | ASH-P@70 [30] | 19.94 | 95.80 | 41.70 | 90.04 | 37.96 | 91.44 | 44.53 | 89.75 | 55.69 | 84.71 | 39.97 | 90.35 |
| | **GAIA-Z (Ours)** | **4.05** | **99.17** | 53.31 | 90.59 | **12.40** | **97.92** | **7.76** | **98.59** | 12.30 | 97.34 | **17.96** | **96.72** |
| | **GAIA-A (Ours)** | 18.34 | 96.51 | **30.98** | **94.54** | _12.73_ | _97.70_ | 16.94 | 96.84 | **14.93** | **97.15** | _18.78_ | _96.55_ |
| CIFAR100 (ResNet34) | MSP [7] | 86.21 | 74.13 | 75.21 | 79.31 | 83.58 | 72.80 | 87.19 | 70.60 | 82.00 | 74.46 | 82.84 | 74.26 |
| | ODIN [9] | 89.34 | 70.21 | 70.00 | 81.44 | 83.80 | 71.37 | 88.10 | 67.69 | 81.81 | 72.66 | 82.61 | 72.67 |
| | Energy [5] | 87.55 | 73.91 | 73.46 | 79.83 | 84.38 | 72.58 | 88.53 | 70.17 | 82.54 | 74.69 | 83.29 | 74.24 |
| | Mahalanobis [14] | 88.71 | 73.72 | 75.70 | 79.57 | 88.28 | 71.63 | 78.54 | 79.74 | 82.63 | 73.78 | 81.29 | 76.16 |
| | ReAct [11] | 77.53 | 83.17 | 71.18 | 78.60 | 73.36 | 84.37 | 78.41 | 80.12 | 72.06 | 82.54 | 74.51 | 81.76 |
| | GradNorm [10] | 90.70 | 65.95 | 80.12 | 61.44 | 82.62 | 58.10 | 92.29 | 64.35 | 83.89 | 52.48 | 86.32 | 60.46 |
| | KNN [29] | 73.34 | 80.06 | 69.24 | 82.17 | 76.98 | 78.36 | 86.76 | 71.53 | 79.95 | 69.24 | 77.25 | 76.27 |
| | Rankfeat [15] | 92.94 | 65.55 | 87.46 | 74.98 | 90.84 | 70.65 | 90.77 | 72.68 | 86.72 | 73.99 | 89.75 | 71.57 |
| | ASH-P@65 [30] | 81.21 | 79.46 | 74.26 | 81.17 | 82.84 | 74.93 | 85.49 | 72.91 | 79.70 | 77.33 | 80.70 | 77.16 |
| | **GAIA-Z (Ours)** | **15.73** | **97.06** | **63.85** | **89.17** | **33.33** | **94.18** | **16.78** | **97.17** | **15.82** | **97.09** | **29.10** | **94.93** |
| | **GAIA-A (Ours)** | _68.02_ | _89.03_ | _68.61_ | _83.33_ | _71.24_ | _86.37_ | _73.15_ | _86.25_ | _63.81_ | _87.12_ | _68.97_ | _86.42_ |
| CIFAR100 (WRN40) | MSP [7] | 83.44 | 79.85 | 76.94 | 77.84 | 76.68 | 80.32 | 85.81 | 72.50 | 83.42 | 74.94 | 81.26 | 77.09 |
| | ODIN [9] | 80.64 | 82.34 | 78.50 | 76.41 | 74.43 | 81.95 | 84.57 | 74.58 | 82.36 | 76.51 | 80.10 | 78.36 |
| | Energy [5] | 84.58 | 79.72 | 76.77 | 77.90 | 76.32 | 80.45 | 86.13 | 72.35 | 83.95 | 74.83 | 81.55 | 77.05 |
| | Mahalanobis [14] | 82.36 | 81.07 | 82.95 | 79.20 | 74.76 | 81.16 | 82.44 | 76.06 | 83.72 | 76.93 | 80.97 | 78.34 |
| | ReAct [11] | 75.04 | 82.36 | 76.09 | 75.83 | 66.64 | 83.06 | 77.94 | 78.18 | 77.66 | 78.33 | 74.67 | 79.55 |
| | GradNorm [10] | 85.27 | 69.22 | 86.58 | 67.75 | 81.10 | 62.38 | 87.01 | 52.89 | 89.41 | 51.30 | 85.89 | 60.71 |
| | KNN [29] | 46.88 | 88.97 | _70.88_ | _82.86_ | 68.92 | 76.83 | 83.57 | 69.64 | 66.13 | 80.39 | | |
| | Rankfeat [15] | 80.39 | 77.10 | 94.58 | 52.35 | 91.63 | 61.89 | 86.83 | 67.71 | 88.00 | 67.36 | 88.29 | 65.28 |
| | ASH-P@70 [30] | 81.20 | 80.99 | 76.24 | 77.92 | 74.78 | 81.06 | 84.81 | 71.73 | 81.97 | 76.12 | 79.80 | 77.97 |
| | **GAIA-Z (Ours)** | **15.19** | **97.19** | 87.06 | 73.42 | _37.97_ | _91.59_ | **25.64** | _95.26_ | **27.29** | **94.05** | **38.63** | _90.30_ |
| | **GAIA-A (Ours)** | _35.49_ | _93.60_ | **53.37** | **89.86** | **33.52** | **93.86** | _27.62_ | **95.37** | _31.44_ | _94.16_ | _36.29_ | **93.37** |

Table 1: **Main Results on CIFAR Benchmarks [7].** We evaluate on ResNet34 [31] and WideRes-net40 [32], which are both pre-trained with cross-entropy loss. For Rankfeat and ASH, we choose their best performance. ↑ indicates larger values are better, while ↓ indicates smaller values are better. All values are percentages. The best results are in Bold and the second best results are underlined.

benchmark are from iNaturalist [23], SUN [33], Places [34] and Textures [35], including fine-grained images, scene-oriented images, and textural images. We also evaluate CIFAR10 and CIFAR100 benchmarks [7], which are routinely used in literature. Correspondingly, OOD datasets are SVHN [36], TinyImageNet [9], LSUN [37], Places [34] and Textures [35].

**Baselines.** We consider various kinds of mainstream post-hoc OOD detection methods as baselines, including Maximum Softmax Probability (MSP) [7], ODIN [9], Energy-based method [5], Maha-lanobis [14], ReAct [11], GradNorm [10], Rankfeat [15], ASH [30] and KNN [29]. We use FPR95 (the false positive rate of OOD examples when the true positive rate of ID examples is 95%) and AUROC (the area under the receiver operating characteristic curve) as evaluation metrics.

## 5.2 Main Results

In our main results, all methods can be directly used for pre-trained models and for a fair comparison, auxiliary OOD data is unavailable for tuning.

**Evaluation on CIFAR benchmarks.** In Tab. 1, we evaluate GAIA methods on CIFAR10 and CIFAR100 benchmarks. The results show that both GAIA-A and GAIA-Z exhibit superior perfor-mance. And we also note that advanced post-hoc methods such as Rankfeat and Gradnorm tend to encounter performance degradations on limited label space with small architectures. For ID dataset CIFAR10, baseline ASH performs the best with an average FPR95 of 26.36% on ResNet34 and ODIN performs 37.09% on WideResNet40 (WRN40). Our method GAIA-Z significantly outper-forms ASH on ResNet34 by **23.10%** improvement and outperforms ODIN on WideResNet by **19.13%** improvement. Moreover, GAIA-A achieves the second best performance after GAIA-Z. For CIFAR100, GAIA-Z attains an average FPR95 of 29.10% and average AUROC of 94.93% on ResNet34, surpassing the best baseline ReAct by a margin of **45.41%** FPR95 and **13.17%** AUROC. GAIA-Z achieves surprising performance on CIFAR benchmarks by utilizing the zero-deflation abnormality.

| Methods Space | Methods | iNaturalist | | SUN | | Places | | Textures | | Average | |
|---|---|---|---|---|---|---|---|---|---|---|---|
| | | FPR95↓ | AUROC↑ | FPR95↓ | AUROC↑ | FPR95↓ | AUROC↑ | FPR95↓ | AUROC↑ | FPR95↓ | AUROC↑ |
| Output | MSP [7] | 63.69 | 87.59 | 79.98 | 78.34 | 81.44 | 76.76 | 82.73 | 74.45 | 76.96 | 79.29 |
| | ODIN [9] | **62.69** | **89.36** | 71.67 | 83.92 | 76.27 | 80.67 | 81.31 | **76.30** | 72.99 | 82.56 |
| | Energy [5] | 64.91 | 88.48 | **65.33** | **85.32** | **73.02** | **81.37** | 80.87 | 75.79 | **71.03** | **82.74** |
| Feature | Mahalanobis [14] | 96.34 | 46.33 | 88.43 | 65.20 | 89.75 | 64.46 | 52.23 | 72.10 | 81.69 | 62.02 |
| | ReAct [11] | 44.52 | 91.81 | 52.71 | 90.16 | 62.66 | 87.83 | 70.73 | 76.85 | 57.66 | 86.67 |
| | KNN [29] | 59.08 | 86.20 | 69.53 | 80.10 | 77.09 | 74.87 | **11.56** | **97.18** | 54.32 | 84.59 |
| | Rankfeat (Block4)[15] | 46.54 | 81.49 | **27.88** | 92.18 | **38.26** | 88.34 | 46.06 | 89.33 | 39.69 | 87.84 |
| | Rankfeat (Block3+4)[15] | 41.31 | 91.91 | 29.27 | **94.07** | 39.34 | **90.93** | 37.29 | 91.70 | 36.80 | 92.15 |
| | ASH-B@90 [30] | **22.22** | **96.15** | 35.43 | 92.53 | 47.73 | 89.61 | 23.33 | 95.43 | **32.18** | **93.43** |
| Gradient | GradNorm [10] | 50.03 | 90.33 | 46.48 | 89.03 | 60.86 | 84.82 | 61.42 | 81.07 | 54.70 | 86.31 |
| | **GAIA-A (Ours)** | **29.47** | **93.52** | **31.24** | **92.42** | **48.55** | **88.94** | 40.41 | 92.71 | **37.42** | **91.90** |
| | **GAIA-Z (Ours)** | 65.09 | 84.15 | 64.23 | 84.31 | 71.02 | 81.16 | **11.32** | **97.93** | 52.92 | 86.89 |

Table 2: **Main Results on ImageNet-1K [28].** OOD detection performance comparison between GAIA and advanced baselines on pre-trained Google BiT-S [38] model. Our methods only use layers from Block4 and all methods are post hoc that can be directly used for pre-trained models. The best results for each Methods Space are all in Bold.

**Evaluation on ImageNet-1K benchmark.** In Tab. 2, we compare GAIA with other post hoc baselines on pre-trained Google BiT-S model [38]. For our methods, both GAIA-A and GAIA-Z use layers from the last block (**Block4**), and no hyperparameters are required. GAIA-A performs well with an average FPR95 of 37.42% and an average AUROC of 91.90%. Compared to other gradient-based OOD methods, GAIA-A outperforms GradNorm by **17.28%** in FPR95. Besides, GAIA-Z excels in handling the OOD dataset of textures with 11.32% FPR95, despite not achieving the best overall performance. While ASH achieves competitive results on the ImageNet dataset through careful parameter tuning, it is highly sensitive to its hyperparameters and lacks empirical parameters. In contrast, GAIA methods don't require parameter adjustments and directly achieve good results.

## 5.3 Ablation Studies

Our ablation study begins by validating the effectiveness of each step of the methods. We first verify the effect of the Frobenius norm (2-norm). Then we explore the aggregation's effectiveness on the label space and the input space.

**Influence of Frobenius norm.** In Eq. 12, we use $\|\mathbf{\Lambda}\|_F$ to calculate the final OOD score. To verify its effectiveness, we evaluate different norms of $\mathbf{\Lambda}$ on the above three benchmarks. As shown in Fig. 4, the Frobenius norm performs the best. Compared to 1-norm, Frobenius norm particularly demonstrates significant improvements. This is because the Frobenius norm can exclude the influence of numerous smaller values. As the number of layers in the model increases, the accumulation of insignificant small values in the shallow layers can weaken the scoring impact of extreme values OOD data. However, we can observe that as the value of $p$ increases, the influence of extreme values will also be affected.

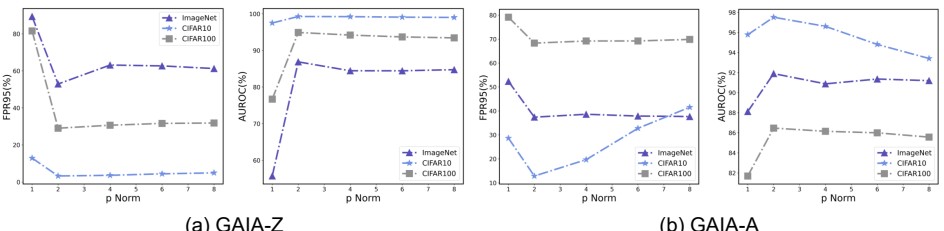

Figure 4: Ablation studies on Frobenius norm of matrix $\mathbf{\Lambda}$.

**Influence of label space aggregation.** In GAIA-A, we employ division to fuse the inner component $\mathbb{E}_{\text{inner}}[\epsilon|\mathbf{A}^{kl}]$ and the output component $\mathbb{E}_{\text{output}}[\epsilon|\mathbf{A}^{kl}]$ to obtain the final OOD scores. As shown in Fig. 5, we visualize the score distributions of the individual components and the fused scores, and observe that the performance of the inner and output components in OOD and ID data are contrasting. After dividing and merging the two components, the fusion resulted in a greater concentration of ID data, tending towards a narrower distribution. However, the impact on the distribution of OOD data was relatively minor, thereby widening the score differences between them. In Tab. 3, we compare the

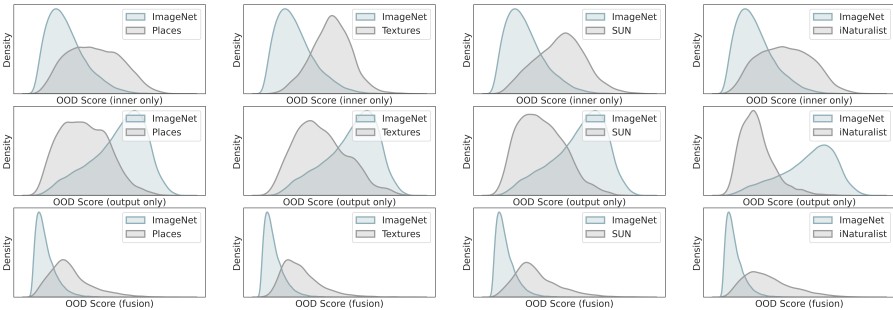

Figure 5: The distribution of the OOD scores in three settings (*inner only*, *output only* and *fusion*). All scores are non-negative for comparison.

OOD detection performance with and without (w/o) the fusion strategy. Experiments demonstrated a improvement with the implementation of this strategy.

| Methods | iNaturalist | | SUN | | Places | | Textures | | Average | |
|---|---|---|---|---|---|---|---|---|---|---|
| | FPR95 ↓ | AUROC ↑ | FPR95 ↓ | AUROC ↑ | FPR95 ↓ | AUROC ↑ | FPR95 ↓ | AUROC ↑ | FPR95 ↓ | AUROC ↑ |
| w/o fusion (top 1 label) | 74.44 | 74.04 | 77.30 | 77.60 | 82.07 | 71.11 | 50.14 | 89.57 | 70.99 | 78.08 |
| w/o fusion (output only) | 47.50 | 92.54 | 69.87 | 84.47 | 74.52 | 82.00 | 76.17 | 78.83 | 67.02 | 84.46 |
| w/o fusion (inner only) | 59.45 | 81.98 | 52.24 | 86.51 | 62.74 | 80.20 | 52.45 | 89.77 | 56.72 | 84.62 |
| **fusion (logsoftmax + division)** | **29.47** | **93.52** | **31.24** | **92.42** | **48.55** | **88.94** | **40.41** | **92.71** | **37.42** | **91.90** |

Table 3: Ablation studies on fusion strategy. *top 1 label* means utilizing the predictive output only.

**Influence of input space aggregation across different layers (blocks).** Given that both ResNet34 and Google BiT-S models have four blocks, we analyze the performance of our methods across different blocks to elucidate the influence of feature layers. As shown in Tab. 4, deeper layers possess a higher power in distinguishing between ID and OOD data. It indicates that as the network becomes shallower, the feature maps progressively contain a diminishing amount of relevant information *w.r.t.* the prediction decision [39]. For CIFAR benchmarks, information from Block3+4 is sufficient for detection, and for ImageNet-1K benchmark, only using Block4 can achieve the best performance.

| Blocks | CIFAR10 | | | | CIFAR100 | | | | ImageNet | | | |
|---|---|---|---|---|---|---|---|---|---|---|---|---|
| | GAIA-A | | GAIA-Z | | GAIA-A | | GAIA-Z | | GAIA-A | | GAIA-Z | |
| | FPR95↓ | AUROC↑ | FPR95↓ | AUROC↑ | FPR95↓ | AUROC↑ | FPR95↓ | AUROC↑ | FPR95↓ | AUROC↑ | FPR95↓ | AUROC↑ |
| Block 1 | 64.52 | 83.28 | 57.15 | 67.06 | 77.89 | 79.72 | 58.38 | 66.46 | 86.59 | 61.26 | 92.38 | 49.62 |
| Block 2 | 62.19 | 86.26 | 50.71 | 86.69 | 77.30 | 80.50 | 52.40 | 86.96 | 87.21 | 58.39 | 88.56 | 58.64 |
| Block 3 | 44.56 | 91.07 | 22.17 | 95.71 | 71.28 | 84.67 | 44.18 | 89.01 | 63.34 | 80.81 | 73.28 | 78.96 |
| Block 4 | 12.90 | 97.54 | 6.42 | 98.75 | 69.16 | 86.40 | 49.97 | 91.28 | **37.42** | **91.90** | **52.92** | **86.89** |
| Block 3+4 | **12.70** | **97.53** | 3.55 | 99.26 | **68.98** | **86.42** | **27.86** | **95.24** | 41.91 | 91.03 | 58.39 | 86.91 |
| All blocks | 12.73 | 97.53 | **3.26** | **99.28** | 68.98 | 86.42 | 29.05 | 94.92 | 42.38 | 90.86 | 63.28 | 86.02 |

Table 4: Ablation studies of the influence on different blocks with average FPR95 and AUROC.

# 6 Related Work

Among all attempts so far, post-hoc methods [5, 9–11, 14, 15] are preferable in the wild due to their advantages of being easy to use without modifying the training procedure and objective. An initial solution proposed by Hendrycks and Gimpel [7] utilizes maximum softmax probability (MSP). While due to the tendency of networks to display overconfident softmax scores when predicting OOD inputs [40, 41], it renders a non-trivial dilemma to separate ID and OOD data. Then ODIN [9] introduces temperature factors and input perturbations to enhance detection performance. In a different approach, Energy [5] is proposed to utilize the energy score as an informative indicator. ReAct [11] proposes that OOD examples result in abnormal model activation and suggests clamping the activation values above a threshold. Rankfeat [15] leverages the differences in singular value distributions, which still focuses on abnormal activations of the model. Another relevant study to this paper is gradient-based OOD detection. In the early work, ODIN [9] first implicitly utilizes gradients as perturbations to increase the softmax score of any given input. Recently, Lee and AlRegib [16], Huang et al. [10] and Igoe et al. [17] use the gradients of parameters as the measurement, which emphasizes the importance of the loss function. In this paper, we delve into investigating attribution abnormality and utilize attribution gradients for OOD detection.

# 7 Discussion

In this section, we discuss the comparison of our methods with other gradient-based OOD detection methods, as well as the limitation on transformer-based models.

## 7.1 Comparison with Other Gradient-based Methods

A crucial distinction between other gradient-based OOD detection methods and ours lies in the utilization of attribution methods to interpret the anomalous behavior of OOD examples. Specifically, we investigate and aggregate the abnormal patterns exhibited by attribution gradients at the feature level. Compared to ODIN [9], GAIA directly leverages the uncertainty derived from the gradients of input features, providing a more intuitive and efficient solution. Furthermore, rather than focusing solely on the softmax output, we delve into the intermediate statistics to uncover more fundamental discrepancies. Compared to GradNorm [10], ExGrad [17] and Lee and AlRegib [16], our approaches focus on attribution gradients and demonstrate superior performance. The comparative performance is presented in Tab. 5. Additionally, GAIA supports batch processing, as the attribution gradients are independent for each input feature, while gradients of parameters are unique to the network. This means that our method can handle multiple samples simultaneously, providing a parallel processing advantage over these methods that can only process one sample at a time.

| Methods | Batch processing | iNaturalist | SUN | Places | Textures | Average |
|---------|:---:|:---:|:---:|:---:|:---:|:---:|
| | | AUROC ↑ | AUROC ↑ | AUROC ↑ | AUROC ↑ | AUROC ↑ |
| Lee and AlRegib [16] | | 72.30 | 82.61 | 74.00 | 84.16 | 78.27 |
| GradNorm [10] | | 90.33 | 89.03 | 84.82 | 81.07 | 86.31 |
| ExGrad [17] | | 76.90 | 66.60 | 68.90 | 65.10 | 69.40 |
| **GAIA-A (Ours)** | ✓ | **93.52** | **92.42** | **88.94** | 92.71 | **91.90** |
| **GAIA-Z (Ours)** | ✓ | 84.15 | 84.31 | 81.16 | **97.93** | 86.89 |

Table 5: Comparison with other gradient-based methods. To ensure a fair comparison with Lee and AlRegib [16], the gradients of uniform noise are used as a surrogate, as suggested in [10].

## 7.2 Limitation on Transformer-based Models

Newer models like Vision Transformers (ViT) [42], which are based on transformers, excel in feature extraction. However, they may not align well with image-specific characteristics. For instance, ViTs employ positional encoding to capture spatial information, posing challenges for attribution. Due to this reason, existing attribution algorithms are rarely applied to ViTs, resulting in poorer performance for GAIA. While the attention mechanism in transformer-based models can also offer directions for visual explanations. In our future work, we will research the uncertainty in the attention matrix to enhance OOD detection performance on transformer-based models.

# 8 Conclusion

This paper targets bridging the gap between OOD detection and visual interpretation by utilizing the uncertainty of a model in explaining its own predictions. We further examine how attribution gradients contribute to uncertain explanation outcomes and introduce two forms of abnormalities for OOD detection. Then, we propose GAIA, a simple and effective framework for abnormality aggregation. The effectiveness of our framework is validated through experiments.

**Societal impact and limitations.** Through this work, we aim to provide a new perspective to improve the performance of OOD detection and ensure the safety and reliability of machine learning applications. However, the utilization of attribution gradients in this paper is relatively simplistic. We believe there is still significant research potential in this area. Moreover, the limitation on transformer-based models remains a topic for further investigation.

# 9 Acknowledgement

Research is supported by the Key Research and Development Program of Guangdong Province (grant No. 2021B0101400003). This work was done while Jinggang Chen was interning at Ping An Technology and the corresponding authors are Xiaoyang Qu and Jianzong Wang from Ping An Technology (Shenzhen) Co., Ltd.

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
