# GAIA: Delving into Gradient-based Attribution Abnormality for Out-of-distribution Detection —-Supplementary Material—-

## A  Extensive Experiments

### A.1  Computational Efficiency of GAIA Methods

Compared to other gradient-based methods, GAIAs support batch processing, as the attribution gradients are independent for each input feature. In Tab. 1, we conduct the test on a Tesla V100 to measure *the average time taken to process a single image* under different batch conditions for both CIFAR benchmarks [1] and ImageNet-1K [2] benchmark.

GAIA methods require the use of attribution gradients from the feature layer of the last block, and the primary time consumption lies in obtaining the attribution gradients through backpropagation. However, as the batch size increases, GAIA experiences accelerated processing since the computations relative to GAIA are comparatively straightforward. With parallelization, a single backward pass can yield attribution gradients for multiple images.

| Settings | MSP [1] | Energy [3] | ODIN [4] | ReAct [5] | GradNorm* [6] | Rankfeat [7] | GAIA-A | GAIA-Z |
|---|---|---|---|---|---|---|---|---|
| CIFAR (Batch=1) | 5.10ms | 5.20ms | 7.23ms | 32.85ms | 25.32ms | 8.85ms | 36.39ms | 35.59ms |
| CIFAR (Batch=128) | 0.24ms | 0.26ms | 0.37ms | 0.73ms | 25.32ms | 3.03ms | 1.01ms | 0.52ms |
| ImageNet (Batch=8) | 49.11ms | 46.03ms | 67.24ms | 59.43ms | 143.47ms | 79.61ms | 54.14ms | 87.24ms |

Table 1: Computational efficiency comparison with other methods. *For GradNorm, the batch size has been consistently set to 1.

### A.2  Deviations with Our Empirical Results

In Tab. 2, we train five ResNet34 models for the CIFAR benchmarks (CIFAR10 and CIFAR100), each using cross-entropy loss under different random seeds. The tests indicate that the performance of GAIA-A and GAIA-Z remains stable, with fluctuation data relative to the average.

| Methods | CIFAR (Avg FPR95) ↓ | CIFAR10 (Avg AUROC) ↑ | CIFAR100 (Avg FPR95) ↓ | CIFAR100 (Avg AUROC) ↑ |
|---|---|---|---|---|
| GAIA-A | 12.73%±2.01% | 97.53%±0.22% | 68.97%±3.13% | 86.42%±1.49% |
| GAIA-Z | 3.26%±1.39% | 99.28%±0.26% | 29.10%±3.36% | 94.93%±0.52% |

Table 2: Additional runs of the method under models trained with differing seeds.

While for the ImageNet-1K benchmark, we use the pre-trained ResNetV2 (BiT) model sourced from the repository of Big Transfer[1]. It is public and most prior methods have evaluated their performance on ImageNet using this pre-trained model as a benchmark. Therefore, we consider this result to be reliable.

---

[1]https://github.com/google-research/big_transfer.

# B  Observations

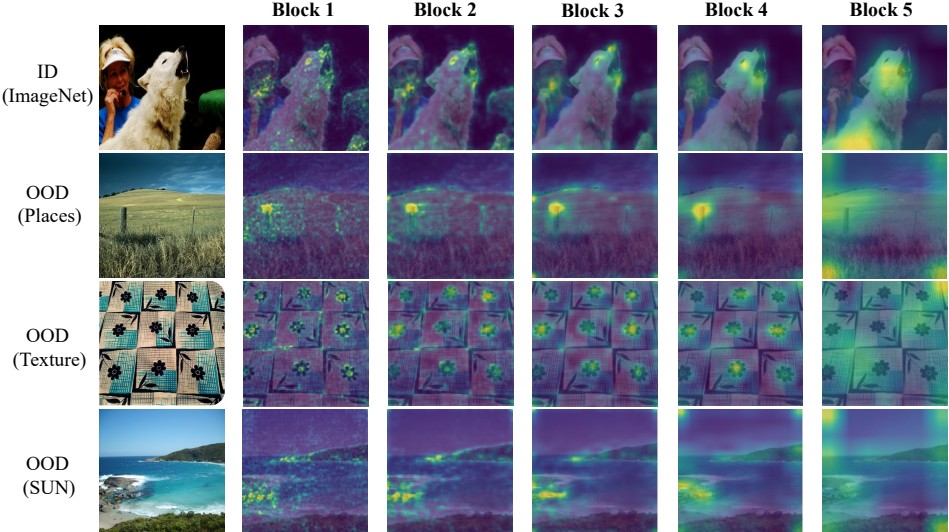

Figure 1: Visualization of the saliency maps from different blocks. We use VGG16 networks (with five blocks) trained on ImageNet [8] and test it on OOD images (Places [9], Texture [10], SUN [11]). The blocks, labeled as block1 to block5, correspond to the output features obtained from shallow to deep.

As shown in Fig. 1, we utilize the LayerCAM [12] (one of the gradient-based CAM methods) to allocate the importance of the output features from different blocks. Compared to the ID image from ImageNet, the saliency maps generated by the model for out-of-distribution (OOD) images from unknown distributions tend to be messy and meaningless. This can be expained as the model's inability to provide reasonable explanations for OOD images within the ID label space. Additionally, at the shallow feature layers, it is observed that the model captures details well for both ID and OOD images. However, as we move towards deeper feature layers, differences emerge between the attributions of ID and OOD images. This indicates that the deep feature layers, which have a greater impact on the model's decision-making process, exhibit better discriminative attributions between ID and OOD images.

# C  Elaboration on Gradient-based Attribution Methods

The gradient with respect to the input features of a neural network is the fundamental piece of information used by several attribution methods to build explanations [12–17]. In this paper, we focus on a line of methods that estimates attribution factor by the product of gradients and input features (*e.g.*, Gradient×Input [15], GradCAM [13], GradCAM++ [14], LayerCAM [12]). Taking GradCAM [13] for example, the importance of each attribution result $M_{ij}$ in the saliency map $\boldsymbol{M} \in \mathbb{R}^{W \times H}$ is defined as:

$$\alpha_{ij} = \text{ReLU}(\sum_{k=1}^{K} w_k A_{ij}^k), \quad \text{where} \quad w_k = \frac{1}{W \times H} \sum_{i=1}^{W} \sum_{j=1}^{H} \frac{\partial S_c(\boldsymbol{A})}{A_{ij}^k} \tag{1}$$

An explanation for CAM methods is that the strength of $A_{ij}^k$ reflects the importance of the different neurons $(i, j, k)$ for the inference of the sample. Moreover, feature maps in different channels have different importance. $w_k$ re-weights them using the average gradient of this channel. To simplify the analysis of GradCAM, we just explain the following attribution $\hat{\alpha}_{ij}$ before the ReLU operation, subject to $\alpha_{ij} = ReLU(\hat{\alpha}_{ij})$. In fact, previous work [13] has proven that gradient-based CAM methods actually explain a DNN as the following linear model of global average pooled feature maps

$\boldsymbol{F}$, where we define $F = [F_1, F_2, ..., F_k]$ and $F_k = \frac{1}{W \times H} \sum_{i=1}^{W} \sum_{j=1}^{H} A_{ij}^k$. Then the output $y_c$ can be explained as:

$$y_c = g(F) = \sum_{k=1}^{K} \underbrace{\frac{\partial S_c(\boldsymbol{F})}{\partial F_k}}_{\text{gradient space}} \cdot \overbrace{F_k}^{\text{feature space}} \tag{2}$$

where $S_c(\cdot)$ represents the output function *w.r.t.* $\boldsymbol{F}$. Gradient-based CAM methods give the explanation for the output of networks with a combination of both attribution gradient space and feature space. Compared to feature space that has been broadly studied in OOD detection, abnormality in attribution gradients are underexplored. In this paper, we focus on how to locate the abnormality in gradient space, which leads to the unreliable importance locations in attribution methods when faced with OOD examples.

## D   Proof of Taylor Explanation in GradCAM

In **Section 4.1**, we introduce channel-wise average abnormality under the assumption that Gradient-based Class Activation Mapping (GradCAM) can be regarded as having only first-order independent terms. Here we provide a proof (from [18]) for this assumption.

*Assumption: Consider $K$-channels feature maps $\boldsymbol{A} \in \mathbb{R}^{K \times W \times H}$ in the convolutional layer as the variables to be attributed. The attribution can be reformulated in form that includes only the first-order Taylor independent terms:*

$$\hat{\alpha}_{ij} = \phi(\boldsymbol{\kappa}) = \sum_{k=1}^{K} \frac{\partial g(\boldsymbol{A})}{\partial A_{ij}^k} A_{ij}^k, \quad \text{where} \quad w_k = \frac{\partial g(\boldsymbol{A})}{\partial A_{ij}^k} = \frac{1}{W \times H} \sum_{i=1}^{W} \sum_{j=1}^{H} \frac{\partial S_c(\boldsymbol{A})}{A_{ij}^k} \tag{3}$$

*where $\boldsymbol{\kappa} = [0, ..., \kappa_i = 1, ...0]$ is a one-hot vector, and $\kappa_j = 0$ if $j \neq i$.*

*Proof:* In GradCAM, $\hat{\alpha}_{ij}$ reflects the total importance of the attribution map $\boldsymbol{A}^{\text{attr}} \in \mathbb{R}^{W \times H}$, where $A_{ij}^{\text{attr}} = \sum_{k=1}^{K} A_{ij}^k$. From the attribution perspective, $\hat{\alpha}_{ij}$ can be estimated by:

$$\hat{\alpha}_{ij} = \sum_{k=1}^{K} \hat{\alpha}_{ij}^k = \sum_{k=1}^{K} \alpha_k A_{ij}^k \tag{4}$$

where $\hat{\alpha}_{ij}^k$ is considered as the attribution of each feature unit $A_{ij}^k$ on the $k$-th channel feature map and $\alpha_k$ represents the average weight for one unit on the feature map. Then based on Eq. 2, we explain the output $y_c$ with explanatory function $g(\boldsymbol{A})$ as an approximate estimation:

$$\begin{aligned} y_c = g(\boldsymbol{A}) &= \sum_{k=1}^{K} (W \times H \cdot \alpha_k) \cdot \boldsymbol{F}^k + \epsilon \\ &= \sum_{k=1}^{K} \alpha_k \sum_{i=1}^{W} \sum_{j=1}^{H} A_{ij}^k + \epsilon \end{aligned} \tag{5}$$

Therefore, we can get:

$$\frac{\partial g(\boldsymbol{A})}{\partial A_{ij}^k} A_{ij}^k = \alpha_k A_{ij}^k = \hat{\alpha}_{ij}^k \tag{6}$$

In Eq. 6, the attribution $\hat{\alpha}_{ij}^k$ can be reformulated as the product of variable $A_{ij}^k$ and the gradient of the explanatory function $g(\boldsymbol{A})$ *w.r.t.* the variable. Following the format of the Taylor expansion mentioned in this paper, we can obtain $\hat{\alpha}_{ij}^k = \phi_k(\boldsymbol{\kappa})$, where $\boldsymbol{\kappa}$ is a one-hot vector that refers to the variable $A_{ij}^k$. It means GradCAM only allocates the first-order independent effect of the $A_{ij}^k$ to its attribution. Furthermore, we consider the variable $A_{ij}^{\text{attr}}$ on the attribution map $\boldsymbol{A}^{\text{attr}}$. Its attribution $\hat{\alpha_{ij}}$ can be reformulated as $\hat{\alpha}_{ij} = \phi(\boldsymbol{\kappa}) = \sum_{k=1}^{K} \phi_k(\boldsymbol{\kappa})$, where $\phi(\boldsymbol{\kappa})$ is the total independent effect wherein $A_{ij}^k$ in each channel $k$ is regarded as one variable $A_{ij}^{\text{attr}}$.

# E  Null-player Axiom

The issue of attribution can be viewed as the assignment of credit in cooperative game theory. This is accomplished by assuming that the network's function serves as a score function, with input features acting as players. Null Player Axiom [19] is proposed to evaluate the accuracy of a particular attribution algorithm. Here, we adopt this concept to describe the quality of attribution on a pre-trained model.

*Null Player Axiom: If removal of a feature across all potential coalitions with other features has no impact on the output, it should be assigned zero importance.*

The feature with zero importance is considered to be *null feature*, which does not contribute to the output score. Therefore, for a null feature, we do not expect any contribution to be assigned in our visual explanation. Formally, there is a set of feature variables $\boldsymbol{Z} = \{z_1, z_2, ..., z_n\}$. One feature $z_i$ is zero-importance for the output $S_c(\cdot)$, if $S_c(\{\boldsymbol{B} \cup z_i\}) = S_c(\{\boldsymbol{B}\})$, where $\boldsymbol{B} \subset \{z_1, z_2, ..., z_n\}/\{z_i\}$.

# F  Analysis of Fusion Strategy

In **Section 4.2**, we introduce the two-stage fusion strategy for GAIA-A and in **Section 5.3**, we briefly discussed the effectiveness of our fusion strategy. Here we provide a more detailed analysis of this fusion strategy.

For GAIA-A, we compute the expectation of channel-wise average abnormality by:

$$\mathbb{E}[\epsilon|\boldsymbol{A}^{kl}] = \frac{\mathbb{E}_{\text{inner}}[\epsilon|\boldsymbol{A}^{kl}]}{\sqrt{\mathbb{E}_{\text{output}}[\epsilon|\boldsymbol{A}_{\text{last}}]}} = \frac{\frac{1}{W \times H} \cdot \|\nabla \boldsymbol{A}^{kl}\|_1}{\sqrt{\frac{1}{W_{\text{last}} \times H_{\text{last}}} \|\nabla \boldsymbol{A}_{\text{last}}\|_1}} \tag{7}$$

where $\mathbb{E}_{\text{inner}}[\epsilon|\boldsymbol{A}^{kl}]$ is considered as the inner component and $\mathbb{E}_{\text{output}}[\epsilon|\boldsymbol{A}_{\text{last}}]$ is considered as the output component.

We first discuss the output component $\mathbb{E}_{\text{output}}[\epsilon|\boldsymbol{A}_{\text{last}}]$. Denotes the output vector as $\boldsymbol{S}(\boldsymbol{A}_{\text{last}}) = [S_1(\boldsymbol{A}_{\text{last}}), S_2(\boldsymbol{A}_{\text{last}}), ..., S_C(\boldsymbol{A}_{\text{last}})] \in \mathbb{R}^{1 \times C}$ (different from the fused score function $\mathcal{S}(\boldsymbol{A}_{\text{last}})$). The output component can be expanded as:

$$
\begin{aligned}
\nabla \boldsymbol{A}_{\text{last}} &= \frac{\partial \mathcal{S}(\boldsymbol{A}_{\text{last}})}{\partial \boldsymbol{A}_{\text{last}}} \\
&= \frac{\partial \sum_{c \in C} \log \text{softmax}(S_c(\boldsymbol{A}_{\text{last}}))}{\partial \boldsymbol{A}_{\text{last}}} \\
&= \frac{\partial \sum_{c=1}^{C} log \frac{e^{S_c(\boldsymbol{A}_{\text{last}})}}{\sum_{i=1}^{C} e^{S_i(\boldsymbol{A}_{\text{last}})}}}{\partial \boldsymbol{A}_{\text{last}}} \\
&= \frac{\partial(\sum_{c=1}^{C} S_c(\boldsymbol{A}_{\text{last}}) - C \cdot log \sum_{i=1}^{C} e^{S_i(\boldsymbol{A}_{\text{last}})})}{\partial \boldsymbol{S}(\boldsymbol{A}_{\text{last}})} \cdot \frac{\partial \boldsymbol{S}(\boldsymbol{A}_{\text{last}})}{\partial \boldsymbol{A}_{\text{last}}} \\
&= [1 - C \cdot \frac{e^{S_1(\boldsymbol{A}_{\text{last}})}}{\sum_{i=1}^{C} e^{S_i(\boldsymbol{A}_{\text{last}})}}, ..., 1 - C \cdot \frac{e^{S_C(\boldsymbol{A}_{\text{last}})}}{\sum_{i=1}^{C} e^{S_i(\boldsymbol{A}_{\text{last}})}}] \cdot \boldsymbol{W}
\end{aligned}
\tag{8}
$$

where $\boldsymbol{W}$ denotes the weight matrix of the final classification layer, which is a constant matrix. From Eq. 8, the effect of output component is similar to the $\boldsymbol{V}$ term in GradNorm [6] when its temperature scaling $T = 1$.

And for inner components, it actually measures the importance of $A_{ij}^{kl}$ at the $l$-th layer to the decision feature map $\boldsymbol{A}_{\text{last}} \in \mathbb{R}^{W_{\text{last}} \times H_{\text{last}}}$:

$$\nabla \boldsymbol{A}^{kl} = \frac{\partial \Phi(\boldsymbol{A}^l, \Theta_\Phi)}{\partial \boldsymbol{A}^{kl}} = \frac{\partial \sum_i^{W_{\text{last}}} \sum_j^{H_{\text{last}}} \Phi_{ij}(\boldsymbol{A}^l, \Theta_\Phi)}{\partial \boldsymbol{A}^{kl}} \tag{9}$$

where $A_{ij}^{\text{last}} = \Phi_{ij}(\boldsymbol{A}^l, \Theta_\Phi)$ and $A_{ij}^{\text{last}} \in \boldsymbol{A}_{\text{last}}$. We compare the inner component with the simple aggregation from output scores without log softmax:

$$\frac{\partial \mathcal{S}(\boldsymbol{A}^l)}{\partial \boldsymbol{A}^{kl}} = \frac{\partial \sum_{c \in C} S_c(\boldsymbol{A}^l)}{\partial \boldsymbol{A}^{kl}} \tag{10}$$

Furthermore, we also investigate the different fusion strategies for $\mathbb{E}[\epsilon|\boldsymbol{A}^{kl}]$ with different factor $q$ in the following equation:

$$\mathbb{E}[\epsilon|\boldsymbol{A}^{kl}] = \frac{\mathbb{E}_{\text{inner}}[\epsilon|\boldsymbol{A}^{kl}]}{\mid \mathbb{E}_{\text{output}}[\epsilon|\boldsymbol{A}_{\text{last}}] \mid^q} \tag{11}$$

The extensive results are shown in Tab. 3. It indicates the effectiveness of our fusion strategy. The visualization of the score distributions is shown in Fig. 2.

| Methods | iNaturalist | | SUN | | Places | | Textures | | Average | |
|---|---|---|---|---|---|---|---|---|---|---|
| | FPR95 ↓ | AUROC ↑ | FPR95 ↓ | AUROC ↑ | FPR95 ↓ | AUROC ↑ | FPR95 ↓ | AUROC ↑ | FPR95 ↓ | AUROC ↑ |
| w/o fusion (top 1 label) | 74.44 | 74.04 | 77.30 | 77.60 | 82.07 | 71.11 | 50.14 | 89.57 | 70.99 | 78.08 |
| w/o fusion (output only) | 47.50 | 92.54 | 69.87 | 84.47 | 74.52 | 82.00 | 76.17 | 78.83 | 67.02 | 84.46 |
| w/o fusion (inner only) | 59.45 | 81.98 | 52.24 | 86.51 | 62.74 | 80.20 | 52.45 | 89.77 | 56.72 | 84.62 |
| fusion (direct from Eq. 10) | 56.41 | 83.13 | 50.14 | 87.31 | 58.63 | 81.77 | 60.66 | 88.06 | 56.46 | 85.06 |
| **fusion (q=0.5)** | **29.47** | 93.52 | **31.24** | **92.42** | **48.55** | **88.94** | **40.41** | **92.71** | **37.42** | **91.90** |
| fusion (q=1) | 30.82 | **94.65** | 49.22 | 91.16 | 60.48 | 87.58 | 52.22 | 90.11 | 48.18 | 90.88 |
| fusion (q=2) | 49.63 | 92.44 | 63.95 | 87.00 | 71.57 | 83.63 | 64.20 | 85.20 | 62.34 | 87.07 |

Table 3: Extensive experiments on fusion strategy.

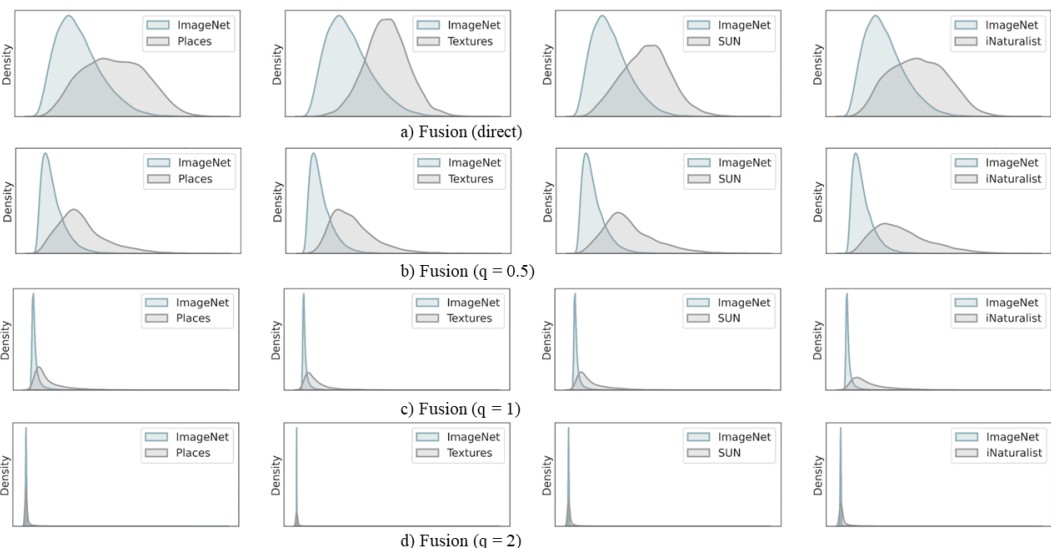

Figure 2: Visualization of the distribution of OOD scores.

## G   Effectiveness of Each Layer in Our Methods

In **Section 5.3**, we studied the influence of input space aggregation across different blocks. The experiments demonstrate that attribution from deeper layers (block 3 and block 4) play a significant role in estimating OOD uncertainty. Fig. 3 illustrates the OOD scores at each layer before aggregation. For CIFAR benchmarks, we evaluate our methods (GAIA-A and GAIA-Z) on ResNet34 with 36 layers. For ImageNet benchmark, we evaluate them on Google BiT-S [20] with 101 layers.

We observed that as the number of layers increases, both the magnitude of the scores and the differentiation between ID and OOD samples gradually increase. For GAIA-A, the scores are primarily concentrated in the final block (block 4) due to the significantly larger absolute values of the mean in those layers compared to the shallow layers. On the other hand, for GAIA-Z, the differences between ID and OOD samples, reflecting matrix sparsity, are more prominent in the deeper layers. As the layers become shallower, the differences tend to decrease.

## H   Experimental Details

Our code is provided in the Supplementary Material. We run our experiments on one Tesla V100 GPU with Python 3.9.15 and Pytorch 1.12.0.

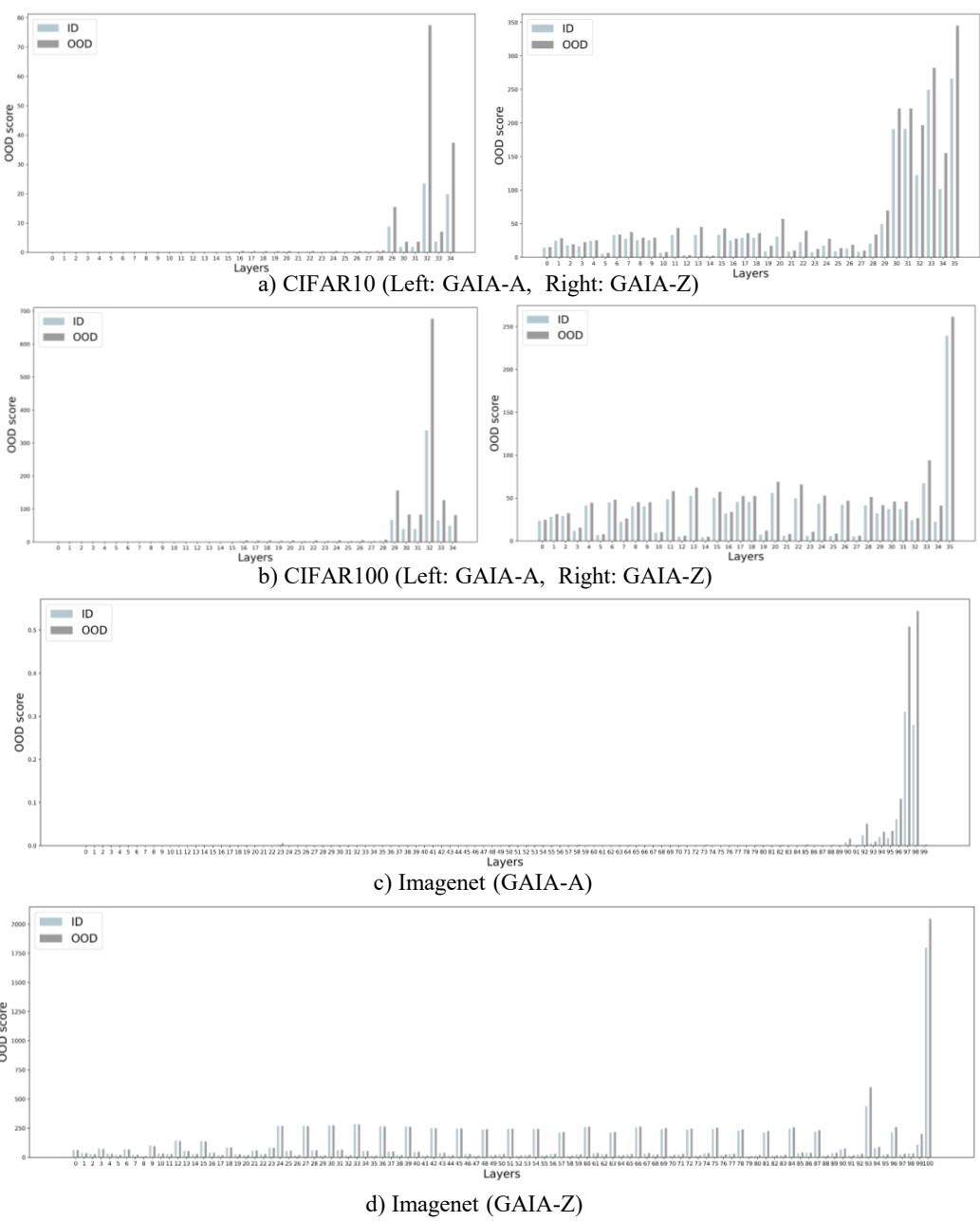

Figure 3: The contribution of each layer to the OOD score.

## H.1    Evaluation Metrics

Following OOD literature [1, 3, 4, 6, 7, 21], we adopt the threshold-free metric [1] to evaluate the performance: FPR95 and AUROC. **FPR95 (FPR at 95% TPR)** can be understood as the likelihood of misclassifying a negative (out-of-distribution) instance as positive (in-distribution) when the true positive rate (TPR) reaches 95%. **AUROC** which stands for the Area Under the Receiver Operating Characteristic curve, represents the overall performance of the classification. The ROC curve illustrates the trade-off between TPR and FPR. The AUROC can be interpreted as the probability that a positive example receives a higher detection score compared to a negative example.

## H.2    Benchmarks

In this paper, we evaluate our methods on two benchmarks: CIFAR benchmarks (CIFAR10 and CIFAR100) [1] and large-scale ImageNet-1K benchmark [2].

### H.2.1    ImageNet-1k Benchmark

As the in-distribution dataset, we utilize ImageNet-1k [2] and conduct evaluations on four out-of-distribution (OOD) test datasets, which are as follows (*All settings remain consistent with previous post-hoc OOD detection studies [1, 3, 4, 6, 7, 21]*):

**iNaturalist** [22] consists of 675,170 training and validation images distributed across 5,089 fine-grained categories found in nature. These categories include major classifications such as plants, insects, birds, and mammals. For evaluation purposes, we randomly select 10,000 images that are distinct from those in the ImageNet-1k dataset.

**Places** [9] consists of 10 million scene images encompassing over 400 distinct scene environments. To evaluate our model's performance, we randomly select 10,000 images that do not overlap with the ImageNet-1k dataset.

**SUN** (The Scene UNderstanding) [11] consists of 397 meticulously sampled categories designed for evaluating scene recognition algorithms. For evaluation purposes, we randomly sample 10,000 images that are distinct from those present in the ImageNet-1k dataset.

**Textures** [10] consists of 5,640 real-world texture images classified into 47 categories. We utilize the entire dataset for evaluation.

### H.2.2    CIFAR Benchmarks

In the literature, CIFAR-10 and CIFAR-100 are commonly employed as in-distribution (ID) datasets. CIFAR-10 consists of 10 classes, while CIFAR-100 consists of 100 classes. We follow the standard split, utilizing 50,000 training images and 10,000 test images. Our approach is evaluated on four widely used out-of-distribution (OOD) datasets, which are listed as follows:

**SVHN** [23] comprises color images depicting house numbers, with ten classes representing digits 0-9. For evaluation, we utilize the entire test set consisting of 26,032 images.

**TinyImagenet** is a subset of the ImageNet dataset [8] and includes 10,000 test images across 200 different classes. We use TinyImageNet (crop) [4], where images are randomly cropped into patches of size 32x32.

**LSUN** [24] contains a testing set of 10,000 images representing 10 different scene categories. Similar to TinyImageNet, we use LSUN (crop) [4], randomly cropping images from the LSUN testing set.

**Places** [9] consists of a vast collection of scene photographs encompassing 365 scene categories. The test set contains 900 images per category. For evaluation, we randomly sample 10,000 images from the test set.

**Textures** [10] comprises 5,640 real-world texture images categorized into 47 classes. We utilize the entire dataset for evaluation purposes.

### H.3 Settings for Baselines

In this paper, we compare our methods with other post-hoc baselines. And for a fair comparison, auxiliary OOD data is unavailable for tuning. Notably, our methods do not require any training or OOD data during uncertainty estimating. Here we summarize the baselines and provide the detailed settings.

**MSP** [1] utilizes the Maximum Softmax Probability (MSP) as a scoring function for OOD detection in one of the early works.

**ODIN** [4] enhances OOD detection through temperature scaling and input perturbation. Following [6], we set the temperature scaling $T = 1000$ in all experiments. For CIFAR10 and CIFAR100, we set perturbation $\epsilon = 0.004$. And for ImageNet, we set $\epsilon = 0$ due to the fact that the addition of perturbation does not further improve the OOD detection performance.

**Energy** [3] introduces the concept of an energy score for OOD detection. To align with the convention where a higher score corresponds to a lower likelihood of being in-distribution, the negative energy score is utilized, which is mapped to a scalar function $\Delta_{\text{Energy}}(\cdot)$

$$\Delta_{\text{Energy}}(x) = -log \sum_{c=1}^{C} exp(S_c(x)) \tag{12}$$

**Mahalanobis distance** [21] employs Mahalanobis distance-based scores for OOD detection. To train the logistic regression model and tune the perturbation magnitude $\epsilon$, we randomly select 500 examples from the in-distribution (ID) datasets and an auxiliary tuning dataset. The tuning dataset consists of adversarial examples generated using the Fast Gradient Sign Method (FGSM) with a perturbation size of 0.05. For ImageNet, we set the $epsilon$ value as 0.005. For CIFAR10 and CIFAR100, we set $\epsilon = 0.001$ and $\epsilon = 0.005$.

**ReAct** [5] is based on the observation that the activations of the penultimate layer exhibit notable differences between ID and OOD data. OOD data tends to trigger very high activations, while ID data follows a well-behaved mean and deviation pattern. In light of this observation, ReAct proposes to clip the activations of the penultimate layer using an upper threshold value $\tau$. To determine $\tau$, we compute the 90th percentile of activations from ID data in the training set. The OOD uncertainty is estimated using the Energy score.

**GradNorm** [6] introduces an approach to estimate OOD uncertainty by leveraging information derived from the gradient space. It computes the Kullback-Leibler (KL) divergence between the Softmax output and a uniform distribution, and then back-propagates the gradient to the last layer. The vector norm of the gradient is directly employed as the scoring function.

**Rankfeat** [7] is built upon the observation that the singular value distributions of in-distribution (ID) and OOD features exhibit notable differences. The OOD feature matrix tends to have a larger dominant singular value compared to the ID feature matrix, and the class predictions of OOD samples are heavily influenced by this singular value. Rankfeat proposes a method that involves removing the rank-1 matrix. In our experiments, we estimate the uncertainty using the output features from the final block.