# OpenReview forum: "GAIA: Delving into Gradient-based Attribution Abnormality for Out-of-distribution Detection"
_NeurIPS.cc/2023/Conference — NeurIPS 2023 poster_

### Official Review · Reviewer_CiZ8 · 2023-06-30

**Soundness:** 3 good
**Presentation:** 3 good
**Contribution:** 2 fair
**Rating:** 5
**Confidence:** 4

**Summary:**

The work looks at leveraging gradient-level attribution information in order to detect semantically shifted OOD samples. In particular, the paper proposes two post-hoc OOD detection methods that leverage the extracted gradient attribution called GAIA-A and GAIA-Z. Both the proposed GAIA-A and GAIA-Z methodologies show strong empirical performance across a wide range of OOD detection tasks.

**Strengths:**

1. The paper provides analysis on an underexplored domain relating to attribution gradient and how to leverage this information for OOD detection.
2. The resulting post-hoc OOD detectors are simple to implement and show strong empirical performance across a wide range of OOD detection tasks.

**Weaknesses:**

There are several other post-hoc methods that have not been included in the empirical evaluation. For example, KNN[1] and Lee and AlRegib [2] would provide fair points of comparison. In addition, if possible, the reviewer would also encourage the authors to include deviations with each empirical result.

[1] Yiyou Sun, Yifei Ming, Xiaojin Zhu, and Yixuan Li. Out-of-distribution detection with deep nearest neighbors. In International Conference on Machine Learning, 2022.

[2] Jinsol Lee and Ghassan AlRegib. Gradients as a measure of uncertainty in neural networks. In 2020 IEEE International Conference on Image Processing (ICIP), pages 2416–2420. IEEE, 2020.

**Questions:**

1. On line 227 the authors hypothesize that "channel-wise average abnormality is better suited for application in scenarios with a large label space." However, the results from CIFAR-100 setting seem to indicate that the choice between GAIA-A and GAIA-Z is not simply based on how large the label space is.
2. On line 151, the authors discuss observations on the number of zero partial derivations for OOD samples. Are there empirical analyses that corroborate these observations?

**Limitations:**

The reviewer would recommend the authors consider adding additional points of comparison as stated in the weakness section above as well as additional runs of the method under models trained with differing seeds.

---

> ### Author Rebuttal · Authors · 2023-08-09
>
> # Response to Reviewer CiZ8
>
> We appreciate your thoughtful review of our work. And we address your questions below:
>
>
> > Q1: Additional runs of the method under models trained with differing seeds.
>
> Thank you for providing valuable suggestions. For the CIFAR benchmarks (CIFAR10 and CIFAR100), **we trained five ResNet34 models, each using cross-entropy loss under different random seeds**. The tests indicate that **the performance of GAIA-A and GAIA-Z remains stable, with fluctuation data relative to the average showcased in Q2**.
>
> The ResNetV2 (BiT) model utilized for ImageNet-1K benchmark is sourced from the repository of Big Transfer [1], which is widely employed as a test model in OOD detection research. Given that retraining is time-consuming, we will include the range of data fluctuation on ImageNet-1K in the paper after conducting the tests. In our paper, **we will add deviations with each empirical result of our methods in the main experiments (both CIFAR and ImageNet-1K benchmarks).**
>
>
> [1] Kolesnikov, Alexander, et al. Big transfer (bit): General visual representation learning. ECCV, 2020.
>
>
> > Q2: Fair comparison with KNN and Lee and AlRegib.
>
> - Comparison with KNN (ResNet34 on CIFAR10 and CIFAR100, ResNetV2 on ImageNet-1K).
>
>
> | Methods | CIFAR10 (Avg FPR95) $\downarrow$ | CIFAR10 (Avg AUROC) $\uparrow$| CIFAR100 (Avg FPR95) $\downarrow$ | CIFAR100 (Avg AUROC) $\uparrow$|ImageNet-1K (Avg FPR95) $\downarrow$ | ImageNet-1K (Avg AUROC) $\uparrow$|
> | ------ | ----| ---- | --- | --- | --- | --- |
> |KNN| 28.14% $\pm$ 1.97%| 95.86% $\pm$ 0.31%| 82.33% $\pm$ 4.87%| 70.21% $\pm$ 8.35%|  53.97% | 85.01% |
> |GAIA-A | 12.73% $\pm$ 2.01% | 97.53% $\pm$ 0.22% | 68.97% $\pm$ 3.13%| 86.42% $\pm$ 1.49% | **37.42%** | **91.90%** |
> |GAIA-Z| **3.26%** $\pm$ 1.39%| **99.28%** $\pm$ 0.26% |  **29.10%** $\pm$ 3.36%| **94.93%** $\pm$ 0.52% | 50.65% | 89.03% |
>
> **In comparison to KNN on the standard-trained network, our method continues to maintain a comprehensive advantage.** An important characteristic of post-hoc OOD detection methods is that they **do not require modifying the training procedure and objective** (all the baselines in our paper are tested on the standard-trained network). Notes that **GAIA-A and GAIA-Z are plug-and-play methods that even do not require in-distribution data for estimation or adjustment of hyperparameters**. Hence, we consider that comparing with KNN on a standard-trained network is fair. Following your suggestion, **we have revised the manuscript to include KNN as a baseline and conducted comparisons in the main experiments.**
>
>
> Nevertheless, we still provide the test results of KNN with contrastive learning for the reviewer's reference. Through comparison, it can be observed that GAIA-A and GAIA-Z perform similarly to the KNN+ method. Especially on the CIFAR10 and CIFAR100 benchmarks, GAIA-Z still outperforms the KNN+. The performance improvement of KNN+ relies on the use of a model trained through contrastive learning, which intervenes in the model's training process. And this could introduce more challenges and uncertainties in practical applications.
>
>
> | Methods | CIFAR10 (Avg FPR95) $\downarrow$ | CIFAR10 (Avg AUROC) $\uparrow$| CIFAR100 (Avg FPR95) $\downarrow$ | CIFAR100 (Avg AUROC) $\uparrow$|ImageNet-1K (Avg FPR95) $\downarrow$ | ImageNet-1K (Avg AUROC) $\uparrow$|
> | ------ | ----| ---- | --- | --- | --- | --- |
> |KNN + contrastive learning| 10.41% | 97.62% | 65.53% | 88.42% | 38.47% | 90.91% |
>
> - Comparison with Lee and AlRegib.
>
> **Please refer to Table 1 in the PDF submitted with the global rebuttal**. Details have been provided in our Supplementary Material (Appendix Section H).
>
> As suggested in GradNorm [1], to ensure a fair comparison, the gradients of uniform noise are used as a surrogate for OOD data during the training of the binary classifier. And our methods outperform Lee and AlRegib. **Regarding this point, we have placed the comparison with other gradient-based methods in the appendix and introduced a  "Discussion" section in the paper to further analyze.**
>
>
> > Q3: GAIA-A and GAIA-Z are not simply based on how large the label space is.
>
> Thank you for your thorough observation. We posit this hypothesis due to that GAIA-A has the ability to aggregate information from all predicted outputs, which demonstrates superior performance on the extensive ImageNet-1K dataset (1000 categories). Furthermore, within the same benchmark, the performance of GAIA-A improves as more labels are aggregated. In Table 5 of the paper, we present the result using only the top-1 label, showcasing a significant performance gap compared to the aggregated results.
>
> We find your perspective to be more rigorous, and we agree that the improvement in GAIA-A's performance on ImageNet is influenced by factors beyond just the expansion of the label space (such as image dimensions, model scale, etc.). **Regarding this point, we have amended these descriptions accordingly. Highlighting the benefits of a broader label space aggregation for GAIA-A aligns with our understanding**, and we will supplement our study with experiments that illustrate how GAIA-A's performance changes on the same benchmark as the aggregated label output quantity increases.
>
>
>
> > Q4: Empirical analyses that corroborate these observations?
>
> **Please refer to Figure 3 in the PDF submitted with the global rebuttal**. For detailed descriptions, please refer to Appendix Section B in the supplementary materials.
>
> In Figure 3, we visualize the sparsity of gradients on feature maps across all channels at a specific layer (measured by the proportion of zero gradients in the entire feature map). Each data point represents the sparsity of one feature map. In deeper layers, OOD images tend to generate attribution gradient matrices with extremely low sparsity across a substantial number of channels, resulting in a remarkable reduction of zero values in the matrix, indicating an abnormal behavior.

---

> ### Comment · Reviewer_CiZ8 · 2023-08-13
> **Response to Author Rebuttal**
>
> The reviewer would like to thank the author for providing the additional experimental evaluations and answering all the existing questions.
>
> **Fair comparison with KNN and Lee and AlRegib**
>
> These experimental results of Lee and AlRegib match with prior expectations and it is encouraging that GAIA is able to outperform all the prior gradient-based and KNN distance-based OOD detection methodologies.
>
> **Other comments and questions**
>
> I thank the reviewer for the clarification and the additional edits. Unfortunately, my general thoughts on the paper remain consistent so I won't raise the review score any higher but I want to encourage the authors to further organize the paper in an effort to improve clarity.

---

> > ### Author Response · Authors · 2023-08-14
> > **Thanks for the comments**
> >
> > We appreciate the response from the reviewer. In our rebuttal, it seems that we have addressed the suggestions and questions raised by the reviewer. Please kindly inform us if the reviewer has any other ongoing concerns or questions?
> >
> > Regarding the issue of the paper's organization, we attach great importance to it. We have already made adjustments and are continuing to refine it further.
> >
> > Our improvements:
> > - We have relocated the "Related Work" section to the end of the manuscript to ensure a smoother flow of our idea.
> > -  In Section 4, we have repositioned Equations 3 and 4 to an additional theoretical analysis section at the end of the paper. Detailed explanations of these equations are provided in the appendix for greater clarity. Moreover, in order to enhance the reader's comprehension of our proposed idea, we have incorporated the visualization that elucidates the connection between the attribution phenomenon and the two abnormalities.
> > - For the ablation experiments, we have restructured the sequence and introduced guiding statements at the beginning to clarify the logical flow of the ablation study. Our ablation study begins by validating the effectiveness of each step of the method, moving from outermost to innermost. Moreover, all Figures and Tables have been arranged coherently, following the sequence from GAIA-A to GAIA-Z.
> > - We have carefully corrected the typos  (grammar, table formatting, descriptive details, mathematical expressions, and so forth).

---

### Official Review · Reviewer_qAeX · 2023-07-02

**Soundness:** 2 fair
**Presentation:** 2 fair
**Contribution:** 2 fair
**Rating:** 5
**Confidence:** 3

**Summary:**

The proposed gradient-based attribution method in this paper is a promising approach that can help distinguish between ID and OOD patterns. By analyzing the uncertainty that arises when models attempt to explain their predictive decisions, the method can provide a more robust and reliable approach to detecting OOD data, which is superior over previous works in gradient-based OOD detection. The authors test their approach on well-known OOD detection benchmarks such as ImageNet and CIFAR, which are widely used in computer vision research. The results demonstrate that the proposed approach is effective in detecting OOD data, outperforming state-of-the-art methods by a significant margin.

**Strengths:**

1. Innovative perspective on quantifying disparities between in-distribution (ID) and out-of-distribution (OOD) data based on analyzing the attribution of embedding features. This approach offers a new perspective on detecting OOD data.

2. Introduces two forms of abnormalities for OOD detection, i.e., the zero-deflation abnormality and the channel-wise average abnormality, which may help to identify OOD data more accurately and effectively.

3. Proposes GAIA, a simple and effective approach that incorporates Gradient Abnormality Inspection and Aggregation, which can be readily applied to pre-trained models without further fine-tuning or additional training.

4. Demonstrates superior performance on both commonly utilized (CIFAR) and large-scale (ImageNet) benchmarks compared to competing approaches, reducing the average FPR95 by 26.75% on CIFAR10 and by 45.41% on CIFAR100.

**Weaknesses:**

1. The authors claim that one can analyze the uncertainty raised when model making decisions via the gradient attribution, and it is the main contribution of this paper. However, I didn't find any theoretical explanation or heuristic justification about how the contribution value is related to uncertainty. With so many systematic analysis in Section 4.1, I think the only key point in supporting why the proposed method works is "Zero-deflation Abnormality" (forgive me if I misunderstand), but it does not explain why the suggest method is a good indicator of prediction uncertainty.

2. The authors claim using the softmax prediction across each class is a novely of their method. However, from my understanding, it is equivalent to the case in using the KL divergence between uniform distribution and model softmax prediction as the objective, which has been well discussed in GradNorm. Therefore, more or less, the author overclaim their contribution.

3. More advanced works, such as ASH, are not compared in the paper. Experimental results with larger models on ImageNet (such as ViT) should also be considered, following previous works such as [1].

4. Since there are some previous works in studying the gradient-based OOD detection, a natural question is why the proposed method is superior over GradNorm. For example, it seems that the authors conduct gradient wrt. model outputs (instead of parameters as in GradNorm), is there any reason for your choice (either heuristically or theoretically); it seems that the authors use the number of non-zero gradients in OOD scoring, what is the superiority to GradNorm in using the L2 norm of gradients.

[1] Yiyou Sun, Yifei Ming, Xiaojin Zhu and Yixuan Li. Out-of-distribution Detection with Deep Nearest Neighbors. ICML, 2022.

**Questions:**

Please refer to the part of Weaknesses

**Limitations:**

Please refer to the part of Weaknesses

---

> ### Author Rebuttal · Authors · 2023-08-09
>
> # Response to Reviewer qAeX
>
> Thank you for your constructive feedback. Before addressing your concerns, we believe there might be **some misconceptions about our method that need clarification**. We will begin by clarifying your certain viewpoints.
>
> > Clarification point 1: **Our methods (both GAIA-A and GAIA-Z) are not based on the gradients with respect to model outputs.**
>
>
> GAIA-A and GAIA-Z are grounded in attribution gradients, specifically **the gradient of the $c$-label output score $S_c(z)$ with respect to the input variable $z$ (i.e., $\frac{\partial S_c(z)}{\partial z}$)**. The term "input variable" generally refers to the input unit or a specific feature unit in the intermediate feature map.
>
> **Attribution gradients are widely utilized in visual explainability techniques** (such as GradCAM, LayerCAM, Integrated Gradients, etc.), and **they are unrelated to the gradients commonly associated with our typical understanding of network optimization (i.e., gradients of the parameters)**. **Hence, our approach fundamentally differs from other gradient-based OOD detection methods**. To provide a better understanding, let's explain the concept of attribution gradients. Attribution gradients refer to the sensitivity of a particular input variable w.r.t. model's predicted output to indicate how the feature influences the model's prediction.
>
>
> > Clarification point 2: About the assertion Weakness 2 raised by the Reviewer.
>
> Our approaches leverage the abnormality in attribution gradients (as mentioned in Clarification Point 1) for OOD detection, **not utilizing the softmax prediction across each class**. Similarly, **our methods are not equivalent to the objective mentioned by the reviewer**. There exists a fundamental distinction between these approaches.
>
> The part that might have caused the reviewer's misunderstanding could be related to the output component of GAIA-A. The core of GAIA-A is centered around utilizing the abnormality in the channel-wise average gradients at certain intermediate layers, which act as visual explanation weights. As we explored the aggregated label space, we discovered that employing log softmax aggregation and splitting it into two parts (output and inner) can further enhance the detection performance. However, it's important to emphasize that **the primary source of enhancement still originates from the inner component**. For more details, please refer to the "Method" section, the "Influence of Label Space Aggregation" in the ablation experiments, and Appendix Section E in the supplementary material.
>
> > Q1: Our main contribution.
>
> **Please refer to the global rebuttal.**
>
> > Q2: not explain why the suggested method is a good indicator of prediction uncertainty.
>
> In this paper, we begin by **addressing an observed phenomenon** (gradient-based attribution methods yield uncertain results on OOD data, **Please refer to Figure 1 in the PDF of global rebuttal**). We then **formulate an explanation for this phenomenon in the context of Taylor expansion (Eq. (3) and Eq. (4))**. Based on the explanation, we **introduce two types of abnormalities to reflect and quantify this uncertainty**, which we derived as effective tools of detecting OOD samples.
>
> GAIA-Z is derived from the Null-player axiom [1], which states that a feature should be considered as having zero importance when it makes no contribution to the model's output. GAIA-Z focuses on determining how certain the model is about its final predictions. On the other hand, GAIA-A places more emphasis on detecting the abnormality arising from gradient-based attribution methods, e.g., GradCAM when they sum the attribution gradients as channel-wise weights (the proof of the relation please refer to appendix section D in the supplementary material). GAIA-A aims to collect extreme outlier values in this process.
>
> [1] Khakzar, Ashkan, et al. "Do explanations explain? Model knows best." CVPR. 2022.
>
> > Q3: Superiority of our methods.
>
> The comparative experiments between them have been included in the PDF of the global rebuttal. Our approach achieves superior performance, offers a fresh perspective, and supports batch processing. (details please refer to Appendix H in the supplementary material)
>
> > Q4: More comparisons and results with larger models.
>
> **The comparison with ASH is shown in the PDF of the global rebuttal  (Table 3)**.
>
> From the data in the Table,  ash_s@90 method slightly outperforms GAIA-A on ImageNet-1K. However, in other conditions, GAIA-A or GAIA-Z performs better. While **ASH achieves competitive results on the ImageNet dataset through careful parameter tuning, it is highly sensitive to its hyperparameters and lacks empirical parameters**. Moreover, these parameters can vary with different model architectures, affecting the practicality of the method. Furthermore, ASH, React, and Rankfeat are similar in that they all rely on deep features of the model. These methods tend to perform well on large datasets like ImageNet but show poorer performance on smaller datasets like CIFAR benchmarks. **In contrast, the GAIA method doesn't require parameter adjustments and directly achieves good results**.
>
> **Following your suggestion, we have revised the manuscript to include ASH as a baseline and conducted comparisons in the main experiments**.
>
> We consider ResNetV2 (BiT) to be a large-scale CNN model. Regarding ViT, our method is not applicable to transformer-based models. ViT employs positional encoding to capture spatial information, posing challenges for attribution (**see Figure 4 of the pdf**). Due to these reasons, existing attribution methods are rarely applied to ViT models, resulting in poorer performance for GAIA on ViT (with an average FPR of 49.13%, compared to the KNN of 38.02%). We acknowledge this limitation of the current GAIA method and have included it in the "Limitations" section. Each proposed method is not without imperfections, and there exists potential for areas of improvement.

---

> > ### Comment · Reviewer_qAeX · 2023-08-19
> > **Thanks for the response.**
> >
> > The authors have addressed my concerns, and I would like to raise my score to 5.

---

> > > ### Author Response · Authors · 2023-08-20
> > > **Thanks for raising the score**
> > >
> > > Thank you! We appreciate the updated score and we are glad that our clarifications have addressed your concerns. Thanks again for taking the time to review our paper and providing detailed comments.

---

### Official Review · Reviewer_kfJE · 2023-07-05

**Soundness:** 3 good
**Presentation:** 3 good
**Contribution:** 2 fair
**Rating:** 5
**Confidence:** 4

**Summary:**

In this paper, the authors propose a novel perspective for quantifying the disparities between in-distribution (ID) and out-of-distribution (OOD) data by analyzing the uncertainty that arises when models attempt to explain their predictive decisions. They investigate the abnormality in gradient-based attribution methods when dealing with OOD data and introduce two forms of abnormalities. They further propose GAIA, a simple and effective approach based on gradient of attribution models for OOD detection. Experimental results demonstrate that GAIA outperforms state-of-the-art methods on CIFAR and ImageNet benchmarks.

**Strengths:**

(1)	The paper offers an innovative perspective on quantifying the disparities between ID and OOD data by analyzing the uncertainties in gradient-based attribution methods, based on zero-deflation abnormality and channel-wise average abnormality.
(2)	The proposed GAIA approach is simple and effective and does not require further fine-tuning or additional training, achieving superior performance to previous SOTAs.

**Weaknesses:**

（1）	The paper lacks clarity and organization in presenting the proposed approach and the experimental results. For example, the mathematical derivation in Section 4 is difficult for readers to follow the flow of ideas. It is unclear to me how Eq. 4 is derived from Eq.3, and what is |·| refer to?
（2）	Ablation studies are comprehensive but messy. It is hard to follow the logic of paper writing. Please give an overview of ablation before the details of each ablation study for better understanding.
（3）	I am curious about the effect of combining GAIA-Z and GAIA-A while no relevant experiment or explanation about it.
（4）	The paper writing should be further improved. There are lots of mistakes. For example, the highlight in Table 2 is wrong (some bests and second bests are upside-down); In line 169 of Page 5, the range of c_i should be placed below the argmax.
（5）	Although the motivation is clear and the proposal is effective, the lack of clarity, implementation details and deficient writing quality are major weaknesses that impact the overall quality of the paper. I recommend you can polish this paper carefully and submit it to other conferences such as cvpr or iclr, which will be good work.

**Questions:**

（1）	I am curious about the effect of combining GAIA-Z and GAIA-A while no relevant experiment or explanation about it.
（2）	The paper writing should be further improved. There are lots of mistakes. For example, the highlight in Table 2 is wrong (some bests and second bests are upside-down); In line 169 of Page 5, the range of c_i should be placed below the argmax.

**Limitations:**

Yes.

---

> ### Author Rebuttal · Authors · 2023-08-07
>
> # Response to Reviewer kfJE
>
> We sincerely appreciate your valuable feedback on our paper and thank you for taking the time to review it. In response to your review, we have addressed the issues raised and made improvements to enhance the clarity, organization, and overall quality of the paper. Below, we outline our rebuttal addressing the specific points you mentioned:
>
> > Concern 1: Clarity and Organization.
>
> Thank you for providing valuable suggestions for improving our paper. We would like to start by outlining the original organization of ideas in the initial manuscript and then proceed to detail the improvements we have made:
>
> Original organization:
>
> In this paper, we begin by addressing an observed phenomenon (gradient-based attribution methods yield cluttered and uncertain results when providing visual explanations for out-of-distribution image prediction by the model). We then formulate an explanation for this phenomenon in the context of Taylor expansion (Eq. (3) and Eq. (4)). Based on the explanation, we introduce two types of anomalies, namely, Zero-deflation abnormality and Channel-wise average abnormality, which we derived as effective tools of detecting out-of-distribution (OOD) samples.
>
> The advantage of this approach is that **it benefits readers' understanding of our idea and the motivation behind our proposed method** (mentioned by Reviewer 1 and Reviewer 2). However, a drawback is that Eq. (3) and Eq. (4) lack sufficient context, which may pose challenges for readers when engaging with the methods.
>
> Our improvements:
>
> - We have relocated the "Related Work" section to the end of the manuscript to ensure a smoother flow of ideas.
> - In Section 4, we have repositioned Equations 3 and 4 to an additional theoretical analysis section at the end of the paper. Detailed explanations of these equations are provided in the appendix for greater clarity. Moreover, in order to enhance the reader's comprehension of our proposed idea, we have incorporated the visualization that elucidates the connection between the attribution phenomenon and the two abnormalities. This visualization expands upon the concept presented in Appendix B by visually highlighting the relationship between the attribution phenomenon of OOD samples and the two abnormalities.
> - For the ablation experiments, we have restructured the sequence and introduced guiding statements at the beginning to elucidate the logical flow of the ablation study. Our ablation study begins by validating the effectiveness of each step of the method, moving from outermost to innermost. We first verify the effect of the Frobenius norm (2-norm), followed by a deeper exploration of the aggregation's effectiveness on the input space (Influence of input space aggregation across different layers (blocks)) and label space (Influence of label space aggregation). Lastly, we validate the overall method's effectiveness across various model capacities. Moreover, all Figures and Tables have been arranged in a coherent manner, following the sequence from GAIA-A to GAIA-Z.
>
> > Concern 2: The effect of combining GAIA-Z and GAIA-A.
>
> This suggestion is highly insightful, as GAIA-A and GAIA-Z exhibit distinct strengths and weaknesses on datasets of different scales. **Combining them presents an interesting idea with potential benefits**.
>
> However, the reason we initially did not explore this avenue in our manuscript was due to that **GAIA-Z and GAIA-A are mutually independent**. They are two approaches that explore model attribution in different directions to reflect the model's uncertainty. GAIA-Z is derived from the Null-player axiom, which states that a feature should be considered as having zero importance when it makes no contribution to the model's output. GAIA-Z focuses on determining how certain the model is about its final predictions. On the other hand, GAIA-A places more emphasis on detecting the abnormality arising from gradient-based attribution methods (e.g., GradCAM) when they sum the attribution gradients as channel-wise weights. GAIA-A aims to collect extreme outlier values in this process.
>
> Therefore, **the magnitudes of the scores generated by these two methods are not directly comparable**. **We attempted a direct summation on benchmarks yet observed no significant enhancement**. The original intention behind proposing GAIA-A and GAIA-Z was to present two plug-and-play methods (not requiring training on in-distribution data) with a focus on minimizing the additional parameters.
>
>
> However, **the direction provided by the reviewer is valuable**, such as normalizing the scores generated by GAIA-A and GAIA-Z using in-distribution data before summation or introducing a tunable or learnable coefficient between the two scores. **We plan to conduct further in-depth research along these lines, incorporating the experimental results in the appendix and highlighting them in future work**.
>
> > Concern 3: Typos in the paper.
>
> We appreciate your attention to the writing quality and errors in the paper. We have dedicated considerable effort to improving the paper's writing. Specifically, we have carefully proofread the manuscript and corrected the typos you mentioned, as well as we have found (grammar, table formatting, descriptive details, mathematical expressions, and so forth).

---

### Official Review · Reviewer_Exp8 · 2023-07-07

**Soundness:** 3 good
**Presentation:** 3 good
**Contribution:** 3 good
**Rating:** 6
**Confidence:** 4

**Summary:**

This paper presents an approach to OOD detection in deep neural networks. The authors propose a method based on analyzing the uncertainty that emerges when models attempt to rationalize their predictive decisions. The abnormalities are found by using two strategies: the zero-deflation abnormality that takes advantage of the observation that attribution gradients in OOD data have more zero values than in-distribution and channel-wise average abnormality that captures variations in the feature maps of OOD data compared to in-distribution data.
The experiments are performed on ImageNet-1K and CIFAR benchmarks and include ablation studies to understand the impact of model capacities.


**Strengths:**

The paper provides an interesting approach to OOD detection by leveraging the concept of attribution gradients. I find the two forms of gradient abnormalities for OOD detection very promising in approaching the problem.

The paper is very well written and clear. The experiments use robust setups on well-known ImageNet-1K and CIFAR benchmarks and the ablation studies are interesting to validate the hypotheses and claims. I suggest experimenting with more datasets to validate the proposition in different contexts.

OOD detection is an important problem in the field of deep learning and has numerous practical applications in enhancing the safety and reliability of deep neural network applications.


**Weaknesses:**

A more direct comparison with other gradient-based methods would be beneficial. While the authors compare their method to GradNorm, it would be interesting to see how GAIA compares to other methods that also utilize gradient information for detection or other tasks.

I think a more detailed explanation of the proposed abnormalities is needed, as it would be beneficial to have a more intuitive or visual explanation to aid in understanding these concepts. The authors could elaborate more in this direction.

While ImageNet-1K and CIFAR are standard benchmarks, it would be important to see the method performs on different types of data, such as text or audio data or even different image datasets.

The paper does not address the computational efficiency and scalability of the proposed method. I think a discussion on the proposed method's potential limitations and failure modes would be a valuable addition.


**Questions:**

Could the authors provide a comparison with other gradient-based methods?

Can the authors elaborate more or provide visual aids to help intuitively understand the proposed abnormalities?

Can the method deal with other types of data (text, audio, etc.)?

Could the authors comment on the computational efficiency and scalability of GAIA?

The authors claim that no hyperparameters are required for their method. Can the authors clarify this point?


**Limitations:**

It is an interesting paper, but a few areas could use more in-depth discussion on the scalability and computational efficiency of the method. While the authors show impressive performance on several benchmark datasets, they do not fully discuss the method's robustness to various forms of OOD shifts, which are common in real-world scenarios.

While both method variants show superior performance compared to other methods, there's a noticeable difference between the two. GAIA-Z generally achieves a lower FPR95 and a higher AUROC than GAIA-A. Is GAIA-Z the more precise variant? Can you elaborate on this direction?

The authors do not discuss where and why the GAIA-A and GAIA-Z methods fail. Failure case analysis is crucial for understanding the limitations of the methods and guiding future research.

---

> ### Author Rebuttal · Authors · 2023-08-09
>
> # Response to Exp8
>
> Thank you for your positive evaluation of our paper. Below, we will address each of your questions:
>
> >Q1: Could the authors provide a comparison with other gradient-based methods?
>
> Regarding the comparative table, **please refer to table 1 in the PDF of the global rebuttal**. Additionally, detailed descriptions and analysis can be found in Appendix Section H of the supplementary materials. **We will include the comparison in a "Discussion" section of the main paper**.
>
> > Q2: Can the authors elaborate more or provide visual aids to help intuitively understand the proposed abnormalities?
>
> **Please refer to figure 1, figure 2, and figure 3 in the PDF of the global rebuttal.**
>
> In Figure 1, we visualize the abnormality in the attributions. Details of figure 2 and figure 3 can be found in Appendix Section B of the supplementary materials.
>
> > Q3: Can the method deal with other types of data (text, audio, etc.)?
>
> Our method is based on attribution gradients, which are extensively utilized in visual interpretation techniques, primarily focusing on the CNN-based network. This line of research also holds significance for certain CNN-based classification tasks. For instance, in audio classification tasks, where GAIA has shown its efficacy (**Please refer to table 2 in the PDF of the global rebuttal**).
>
>
>
> > Q4: Could the authors comment on the computational efficiency and scalability of GAIA?
>
> Compared with other gradient-based methods, GAIAs support batch processing, as the attribution gradients are independent for each input feature.
>
> We conducted tests on a Tesla V100 to measure the average time taken to process a single image under different batch conditions for both CIFAR benchmarks and the ImageNet benchmark.
>
> | Settings | MSP | Energy | ODIN | ReAct | GradNorm* | RankFeat | GAIA-A | GAIA-Z|
> | ------ | ----| ---- | --- | --- | --- | --- | --- | --- |
> |CIFAR(batch=1) | 5.10ms | 5.2ms | 7.23ms | 32.85ms |  25.32ms | 8.85ms | 36.39ms | 35.59ms |
> |CIFAR(batch=128) | 0.24ms | 0.26ms|  0.37ms | 0.73ms | 25.32ms| 3.03ms | 1.01ms | 0.52ms |
> |ImageNet(batch=8) | 49.11ms | 46.03ms | 67.24ms | 59.43ms | 143.47ms | 79.61ms | 54.14ms | 87.24ms |
>
> ***For GradNorm, the batch size has been consistently set to 1.**
>
> GAIA methods require the use of attribution gradients from the feature layer of the last block, and the primary time consumption lies in obtaining the attribution gradients through backpropagation. However, **as the batch size increases, GAIA experiences accelerated processing since the computations relative to GAIA are comparatively straightforward. With parallelization, a single backward pass can yield attribution gradients for multiple images**.
>
> > Q5: The authors claim that no hyperparameters are required for their method. Can the authors clarify this point?
>
>
> GAIA-A and GAIA-Z are plug-and-play OOD detection methods, meaning they do not have hyperparameters that need adjustment on different in-distribution datasets. For example, methods like Energy, ODIN, ReAct, and Mahalanobis require tuning their hyperparameters (temperature, perturbation, thresholds, etc.) for different in-distribution datasets.
>
> > Q6: Discuss the method's robustness to various forms of OOD shifts.
>
> This is a valuable suggestion. We have also taken into consideration various shift scenarios.
>
> Specifically, we explore whether our approach can still categorize images as ID when there is a domain shift (covariate shift). We employed CIFAR-10C as the dataset exhibiting domain shift and compared its scores with those of CIFAR-10. The similarity in scores between the two datasets indicates the robustness of our method in handling in-distribution test data affected by domain shift.
>
>
> > Q7: Is GAIA-Z the more precise variant? Can you elaborate on this direction?
>
> We consider that GAIA-A is also an effective method. GAIA-Z performs well when dealing with small-scale images, such as the CIFAR benchmarks. However, when applied to larger datasets like ImageNet, it may encounter more significant disturbances and challenges. On the other hand, GAIA-A has the advantage of collecting more anomalies in larger label spaces, making it more effective on datasets like ImageNet. Additionally, the two-stage enhancement process further improves its performance on the ImageNet dataset. Moreover, GAIA-A's ability to detect OOD through analyzing anomalous behavior in visual attribution methods provides insightful implications and holds potential for further exploration in this direction.
>
> > Q8: Where and why the GAIA-A and GAIA-Z methods fail.
>
>  Newer models like Vision Transformers (ViT), which are based on transformers, excel in feature extraction. However, they may not align well with image-specific characteristics. For instance, ViT employs positional encoding to capture spatial information, posing challenges for attribution. Due to these reasons, existing attribution methods are rarely applied to ViT models (**see figure 4 of the pdf**), resulting in poorer performance for GAIA on ViT. **We acknowledge this limitation of the current GAIA method and have included it in the "Limitations" section. Furthermore, it serves as a potential avenue for future improvements and exploration.**

---

> > ### Comment · Reviewer_Exp8 · 2023-08-16
> >
> > Thank you for providing a comprehensive rebuttal and the effort to improve the paper.
> >
> > The visualizations incorporated offer a clearer understanding of the abnormalities. Looking at the images, I was also considering if a signal-to-noise analysis could provide some insights if the features considered are more related to the foreground than the background.
> >
> > Also, considering your last comment, I'm not sure I understand the problem of the proposed approach to deal with ViT. Given the growing popularity and effectiveness of transformer architectures, how to adapt or evolve the proposed to accommodate these new advancements?

---

> > > ### Author Response · Authors · 2023-08-17
> > >
> > > We appreciate your insightful viewpoints and thoroughly enjoy the discussion.
> > >
> > > > Q1: Looking at the images, I was also considering if a signal-to-noise analysis could provide some insights if the features considered are more related to the foreground than the background.
> > >
> > > This is indeed an interesting perspective. We have given consideration to this issue and have come up with a few directions to explore.
> > >
> > >
> > > If we consider the attribution map on the feature map as a grayscale image, with attribution values as pixel values, features deemed to contribute to predictions (useful features) could be considered as signals.
> > >
> > > Under the assumption of the highly effective visual explanation method, these useful features, such as the foreground, should have high pixel values, making them meaningful signals. On the other hand, the background should possess low pixel values, representing insignificant noise. Therefore, we can use the signal-to-noise ratio (SNR) to quantify the uncertainty of the model's attribution explanations:
> > >
> > > $$\text{SNR} (
> > >     \text{db})= \frac{\text{Signal Power}}{\text{Noise Power}}$$
> > >
> > > If the model makes a confident and correct decision, the SNR should be very high. However, this approach also presents some challenges: 1) How do we define the scope of the signal (foreground). We may need a threshold or algorithm to differentiate which regions on the attribution map are background and which are foreground. 2) When there are no in-distribution objects in the entire image (OOD), we should consider the entire image as noise. In our observations, this scenario can result in widely scattered and exceptionally high noise values. We need to design a more appropriate matrix to reflect SNR.
> > >
> > > **We consider that this approach could be highly beneficial for object-level OOD detection.** When an image contains multiple objects that need to be recognized, we can consider the regions within the bounding boxes as the signal (foreground) and those outside the boxes as the background. Then we can calculate the SNR on the attribution map to reflect the model's uncertainty about predictions within these regions. This method might be effective in detection scenarios with relatively stable environments, such as autonomous driving, industrial identification, etc.
> > >
> > >
> > >
> > > > Q2: Considering your last comment, I'm not sure I understand the problem of the proposed approach to deal with ViT. Given the growing popularity and effectiveness of transformer architectures, how to adapt or evolve the proposed to accommodate these new advancements?
> > >
> > > Given the current trends, providing visual explanations for large models has been a focal point for both the interpretability academic community and the engineering efforts in recent years. Honestly, this is also the research direction we are currently pursuing.
> > >
> > > Overall, our approach involves utilizing model-explained prediction uncertainty for OOD detection. For large models, our ongoing research primarily focuses on two directions:
> > >
> > > - Direction 1: **For transformer-based model.**
> > >
> > > Despite the differences between transformer-based architectures and the convolutional feature layers of CNNs, methods based on attribution gradients can still provide explanations for predictions made by models using transformer structures (such as ViT). For the transformer-based model, we consider the attribution gradients on the attention matrix.
> > >
> > > Many traditional CAM methods, including GradCAM, are initially proposed based on CNN structures, and therefore, their performance on transformer-based models might be suboptimal. **However, there is a line of improvements available now to enhance the gradient-based visual explanations on such models**. One prominent example is [1]. Building upon these enhancements, we are currently researching how to identify attribution gradient abnormality on the attention matrix to reflect the model's uncertainty.
> > >
> > >
> > > Furthermore, the Attention mechanism in transformer-based models can also offer directions for visual explanations. Researching uncertainty in this context can further enhance OOD  detection.
> > >
> > >
> > > [1] Chefer, Hila, Shir Gur, and Lior Wolf. Transformer interpretability beyond attention visualization. CVPR. 2021.
> > >
> > > - Direction 2: **For CNN-based backbone.**
> > >
> > > Although the mainstream of large models isn't purely CNN-based, many multimodal large models (like CLIP) and downstream tasks (such as object detection) still utilize CNNs as the backbone networks for visual feature extraction.
> > >
> > > How to reflect the model's uncertainty in visual feature extraction on these backbones is also a topic we are currently researching.

---

> > > > ### Comment · Reviewer_Exp8 · 2023-08-18
> > > >
> > > > I appreciate your answer, clarification, and additional edits. I think the review score is still consistent. Congratulations on the work.

---

> > > > > ### Author Response · Authors · 2023-08-20
> > > > > **Thanks for your positive feedback**
> > > > >
> > > > > Thanks for your positive feedback and insightful comments! We are delighted that our responses have been satisfactory, and we thoroughly enjoy the discussions with you!

---

### Official Review · Reviewer_TjRi · 2023-07-27

**Soundness:** 2 fair
**Presentation:** 3 good
**Contribution:** 2 fair
**Rating:** 5
**Confidence:** 4

**Summary:**

In this paper, the authors present an innovative perspective on quantifying the disparities between in-distribution (ID) and out-of-distribution (OOD) data. The authors observed that gradient-based attribution methods face challenges when assigning feature importance to OOD data, leading to significantly divergent explanation patterns.

To address this issue, the authors investigate how attribution gradients contribute to uncertain explanation outcomes and introduce two forms of abnormalities for OOD detection: the zero-deflation abnormality and the channel-wise average abnormality. To overcome these challenges, they propose a new approach called GAIA (Gradient Abnormality Inspection and Aggregation), which is simple yet effective. Importantly, GAIA can be directly applied to pre-trained models without the need for further fine-tuning or additional training. The results demonstrate that GAIA outperforms existing approaches on commonly utilized benchmarks such as CIFAR and large-scale benchmarks like ImageNet. Specifically, on CIFAR benchmarks, GAIA reduces the average FPR95 by 26.75% on CIFAR10 and by 45.41% on CIFAR100 when compared to competing methods, highlighting its superiority in OOD detection.

**Strengths:**

- The idea of the paper is clear, the writing is easy to follow, and provides theoretical support.
- The proposed method GAIA is simple and effective on CIFAR benchmarks.

**Weaknesses:**

But I'm more concerned about the effectiveness of the method:

- In the comparison in Table 1 on large-scale benchmark ImageNet-1K, the proposed GAIA-A and GAIA-Z methods only compare the inferior version of Rankfeat (Block 4), but don't outperform the SOTA version of Rankfeat (Block 3 + 4)$[1]$.

- There are two versions of GAIA: GAIA-A, and GAIA-Z, which have their own strengths and weaknesses on different benchmarks, but there is no guidance in the paper on which method to use in different benchmarks.

- Important points of innovation and the bulk of proofs are related to GAIA-A, but GAIA-A does not outperform GAIA-Z on most benchmarks (e.g. CIFAR). This reduces the effectiveness of GAIA-A.

- Typo: line 173, "with with".

Reference:

[1] Song Y, Sebe N, Wang W. Rankfeat: Rank-1 feature removal for out-of-distribution detection[J]. Advances in Neural Information Processing Systems, 2022, 35: 17885-17898.

**Questions:**

1. What is the meaning of zero baseline output S_c ($0$) in Eq. (3)?
2. What are the inherent reasons for the difference in effectiveness between GAIA-A and GAIA-Z?

For more please refer to the Weaknesses part.

Open question:
- Pre-training on one dataset and testing on another, yet the dataset itself is natural domain inconsistent, is there a real-world application for this?

- What is the OOD sample in the real-world application? If we only have the dataset on a sunny day, but a bird on a rainy day, would that be considered as an OOD sample? Will the method GAIA in the paper recognize birds in rainy weather as OOD samples? Or only images that do not contain objects will be treated as OOD?

**Limitations:**

Yes

---

> ### Author Rebuttal · Authors · 2023-08-07
>
> # Response to Reviewer TjRi
>
> Thank you for your constructive feedback. We are pleased that you find our presentation clear and easy to follow. And we address your questions below:
>
>
> **For the open questions, we have included the replies in the global rebuttal due to the limit constraints of the rebuttal (not exceeding 6000 characters).**
>
> > Q1: What is the meaning of zero baseline output $S_c (0)$ in Eq. (3)?
>
> In Eq. (3), "baseline" can be understood as **the initial value used as a reference in attribution methods**. This analytical form is commonly adopted in visual interpretability methods. In our paper, we consider the feature values to be all zeros as the "zero baseline", which is commonly adopted for analyzing gradient-based attribution methods. $S_c(0)$ represents the model's c-label output w.r.t baseline. Then in Eq. (4), $|S_c(z) - S_c(0)|$ represents the c-label output change caused by feature $z$ in the model's predictions. Using the zero baseline also simplifies the form of the Taylor expansion, making it easier for further analysis. With this approach, the output change in Eq 4 can be represented as a combination of feature gradients with respect to the output. Consequently, it becomes possible to determine the contribution of each feature gradient to the final output change.
>
> > Q2: What are the inherent reasons for the difference in effectiveness between GAIA-A and GAIA-Z?
>
> In this paper, **we target bridging the gap between OOD detection and visual interpretation by utilizing the uncertainty of a model in explaining its own predictions**. GAIA-Z and GAIA-A are two approaches that explore model attribution in different directions to reflect the model's uncertainty. GAIA-Z is derived from the Null-player axiom [1], which states that a feature should be considered as having zero importance when it makes no contribution to the model's output. For example, if the model makes overconfident predictions for OOD samples (e.g., classifying the grassland as a bird), **GAIA-Z focuses on determining how certain the model is about its final predictions**. In contrast, when using visual interpretation to explain why a sample is classified as a bird, GAIA-Z might produce many non-zero importance features, leading to messy attribution maps. On the other hand, **GAIA-A places more emphasis on detecting the abnormality arising from gradient-based attribution methods** (e.g., GradCAM when they sum the attribution gradients as channel-wise weights. GAIA-A aims to collect extreme outlier values in this process.
>
>
> [1] Khakzar, Ashkan, et al. "Do explanations explain? Model knows best." CVPR. 2022.
>
>
>
> > Q3: No guidance in the paper on which method to use in different benchmarks;
>
> GAIA-Z performs well when dealing with small-scale images, such as the CIFAR benchmarks. However, when applied to larger datasets like ImageNet, it may encounter more significant disturbances and challenges. On the other hand, GAIA-A has the advantage of collecting more anomalies in larger label spaces, making it more effective on datasets like ImageNet. Additionally, the two-stage enhancement process further improves its performance on the ImageNet dataset.
>
>
>
>
> > Q4: Comparision with Rankfeat (Block 3 + 4).
>
> Thanks for your comments. We compared our method with Rankfeat (block 4) because **our approaches also only utilize information from block 4**. When using the same amount of information, GAIA-A achieves better results. Additionally, **including data from an extra block would lead to a decrease in inference time performance**. In our testing on CIFAR10 (batch size 128), Rankfeat (block3+4) takes an average of 5.3ms to process a single image, while GAIA-A takes an average of 1.01ms and GAIA-Z takes an average of 0.52ms. In this case, GAIA is at least five times faster. **Surely, to provide a more objective comparison, we have modified Table 1 and Table 2 to add data from RankFeat (block3+4) as a reference**. While RankFeat (with an average FPR95 of 36.80%) performs slightly better than GAIA-A (with an average FPR95 of 37.42%) on ImageNet, **GAIA still demonstrates significant improvements on CIFAR benchmarks**.
>
> | Methods | CIFAR10 (Avg FPR95) $\downarrow$ | CIFAR10 (Avg AUROC) $\uparrow$| CIFAR100 (Avg FPR95) $\downarrow$ | CIFAR100 (Avg AUROC) $\uparrow$|ImageNet-1K (Avg FPR95) $\downarrow$ | ImageNet-1K (Avg AUROC) $\uparrow$|
> | ------ | ----| ---- | --- | --- | --- | --- |
> |RankFeat (3+4)| 62.46% | 83.62% | 90.75% | 68.99% | **36.80%** | **92.15%** |
> |GAIA-A | 12.73% | 97.53% | 68.97% | 86.42% | 37.42% | 91.90% |
> |GAIA-Z| **3.26%** | **99.28%** |  **29.10%** | **94.93%** | 50.65% | 89.03% |
>
>
> > Q5:  Proof in GAIA-Z less than GAIA-A and the effectiveness of GAIA-A.
>
> As mentioned in Q2, GAIA-Z, and GAIA-A both provide valuable insights into model uncertainty, offering a way to connect OOD detection and visual explanation methods. **They are mutually independent, and each method holds its value and insights**.
>
>
> We have conducted a motivational analysis of the zero-deflation abnormality in section 4. In response to the reviewer's suggestion (analysis for GAIA-Z is less extensive compared to GAIA-A), **we included further theoretical analysis of GAIA-Z based on the Null-player axiom (mentioned in Q2) in the appendix**.
>
>
> **We consider that GAIA-A is also an effective method**. On the ImageNet-1k benchmark, GAIA-A performs better than GAIA-Z, indicating that GAIA-A has its own advantages over other gradient-based methods on larger datasets. ImageNet-1K is a large-scale dataset comparable to CIFAR benchmarks in terms of scale. Moreover, GAIA-A's ability to detect OOD through analyzing anomalous behavior in visual attribution methods provides insightful implications and holds potential for further exploration in this direction.
>
> > Q6： Typo: line 173, "with with."
>
> We sincerely appreciate your thoroughness and attention to detail. We have diligently reviewed and corrected the typos in our manuscript.

---

> ### Comment · Reviewer_TjRi · 2023-08-14
> **Response to Authors**
>
> The authors addressed my concerns to some extent. Could the authors provide a more professional analysis of why "GAIA-A has the advantage of collecting more anomalies in larger label spaces"? Not just from an experimental observation view.

---

> > ### Author Response · Authors · 2023-08-15
> > **Thanks for your comments**
> >
> > Thank you for taking the time to read our rebuttal and engaging in timely discussions with us.
> >
> > > Q: Could the authors provide a more professional analysis of why "GAIA-A has the advantage of collecting more anomalies in larger label spaces"? Not just from an experimental observation view.
> >
> > Of course! We put forward this viewpoint due to that GAIA-A has the ability to aggregate information from all predicted outputs.
> >
> > As mentioned in the rebuttal, GAIA-A aims to gather extreme anomaly values of weights (channel-wise average gradients) in the gradient-based attribution method to reflect uncertainty. Consider that the aggregation region has $L$ feature layers, and each layer has $K$ channels for ease of representation. The overall expectation of the abnormality $\mathbb{E}[\epsilon]$ can be represented as:
> >
> > $\begin{equation}
> >     \mathbb{E}[\epsilon] =  \sqrt{\sum\limits_{l\in L} \sum\limits_{k\in K} \| \mathbb{E}[\epsilon|\textbf{A}^{kl}]\|^2}
> > \end{equation}$
> >
> > Then, we analyze the expectation of the abnormality on an individual k-channel,
> >
> > $\begin{equation}
> >     \mathbb{E}[\epsilon|\textbf{A}^{kl}] = \left\|  \sum\limits_{i, j} \frac{\partial \Gamma(S_c(\textbf{A}^l))}{\partial \textbf{A}^{kl}_{ij}} \right\| = \left\| \Omega \right\|
> > \end{equation}$
> >
> > where $\Gamma(\cdot)$ represents a method of aggregating over label outputs, while $\Omega$ signifies the aggregation of attribution gradients. In our paper, we utilize the log-softmax aggregation approach. For the sake of simplicity in analysis, let us consider aggregation as a summation. Thus, the aggregation gradient $\Omega$  can be decomposed as follows:
> >
> > $\begin{equation}
> >     \Omega = \frac{\partial \sum\limits_{c\in C}S_c(\textbf{A}^l)}{\partial \textbf{A}^{kl}} = \sum\limits_{c\in C} \frac{\partial S_c(\textbf{A}^l)}{\partial \textbf{A}^{kl}} = \sum\limits_{c\in C} \omega_c
> > \end{equation}$
> >
> > where $w_c$ represents the weight corresponding to the attribution for the $c$-label output $y_c$. Therefore, the expectation of abnormality on this channel can be represented as the sum of the expectation of abnormality across the whole label space.
> >
> > $\begin{equation}
> >     \mathbb{E}[\epsilon|\textbf{A}^{kl}] = \sum\limits_{c\in C} \mathbb{E}[\epsilon|\textbf{A}^{kl}, y_c]
> > \end{equation}$
> >
> > In other words, since GAIA-A relies on collecting weight anomalies, a larger label space allows it to gather more anomalies, thereby better reflecting the model's uncertainty.

---

> > > ### Comment · Reviewer_TjRi · 2023-08-18
> > > **Response to Authors**
> > >
> > > Thank you for addressing each of my concerns. I will rate this paper as borderline acceptable.

---

> > > > ### Author Response · Authors · 2023-08-20
> > > > **Thanks for raising the score**
> > > >
> > > > Thank you for raising the score. We are delighted that our responses were able to address your concerns. Finally, we appreciate your careful review and engaging discussions.

---

### Author Rebuttal · Authors · 2023-08-09

We appreciate all the reviewers' time and valuable feedback. We are delighted that the reviewers found our article to be **clear**, **easy to read** (**R1**, **R2**), and regarded our method as both **simple and effective (R1, R3, R4, R5)**. It is also great to hear that our findings are **interesting and innovative (R2, R3, R4)**.

We have addressed the reviewers' comments and concerns in individual responses to each reviewer. And we have summarized the changes as follows:

- We have added comparisons with RankFeat (block 3+4) (**R1**), ASH (**R4**) and KNN (**R5**) in our experiments.
- We have included further theoretical analysis of GAIA-Z based on the Null-player axiom in the appendix (**R1, R4**).
- We have introduced a new "Discussion" section to analyze the differences between our method and other gradient-based approaches (**R2, R4, R5**).
- We have discussed the limitations of GAIA on transformer-based models in the "Limitations" section (**R2, R4**).
- We have restructured the presentation of the "Method" section to enhance understanding (**R3**).
- We have dedicated considerable effort to improving the paper's writing (**R3**) and have corrected the typos (**R1, R3**).
- We have rephrased our assumptions regarding performance on a large label space in a more rigorous manner and will supplement our study with experiments (**R5**).
- We plan to conduct further in-depth research on the effect of combining GAIA-Z and GAIA-A (**R3**).
- We will add deviations with each empirical result of our methods in the main experiments (**R5**).

***R1:** TjRi, **R2:** Exp8, **R3:** kfJE, **R4:** qAeX, **R5:** CiZ8.

**Our main contribution**:

Our main contribution is that we target **bridging the gap between OOD detection and visual interpretation by utilizing the uncertainty of a model in explaining its own predictions**. Visual explainability methods are employed to attribute a model's predictive outcomes. We endeavor to uncover uncertainty when explaining anomalies outside the label space, aiming to detect OOD samples. This is a novel domain, as the realm of visual interpretability based on attribution gradients is vast and theoretically comprehensive. We believe this constitutes a highly promising avenue for research.

Below are the responses to the open questions raised by R1:

>Open question 1: Pre-training on one dataset and testing on another, yet the dataset itself is naturally domain inconsistent, is there a real-world application for this?

OOD detection aims to ensure the trustworthiness and safety of machine learning models in an open-world setting. In practical deployment, pre-trained models may encounter unknown natural inputs that surpass their cognitive capabilities, leading to overconfident decision-making. For instance, for a trained food classifier, when a user uploads a non-food image, we hope to have a method that can recognize this as an unknown input and refrain from misclassifying it into any food category erroneously. In safety-critical scenarios like autonomous driving systems, when the driving system identifies unknown objects, it should trigger an alert and hand over control to the driver.



>Open question 2: What is the OOD sample in the real-world application? If we only have the dataset on a sunny day but a bird on a rainy day, would that be considered an OOD sample? Will the method GAIA in the paper recognize birds in rainy weather as OOD samples? Or only images that do not contain objects will be treated as OOD?

OOD detection aims to ensure safety during deployment by identifying natural samples that the model was not originally trained to recognize. For instance, if the model is trained to classify animal categories such as birds, dogs, and cats (ID), then airplanes can be considered as OOD samples (called semantic shifts). If the training data includes images of birds on a sunny day, the images of birds on a rainy day will be considered from a different domain. This is typically referred to as covariate shift. In such cases, researchers often focus on the model's transferability or generalization capabilities, which are separate research topics (such as domain adaptation, domain generalization, etc). In the benchmarks we use, the testing and training ID samples are kept separate. Taking the CIFAR10 benchmark as an example, the model is trained on the training set, but during the evaluation, the model is tested on an unseen testing set, which includes birds in different environments. Our method is capable of effectively distinguishing between ID samples (that the model has seen during training) and OOD samples (that are novel to the model).

---

> ### Comment · Area_Chair_S8dB · 2023-08-13
> **Discussion starts**
>
> Dear Reviewers,
>
> Thank you for reviewing this paper. Authors have provided their rebuttal. Would you please check it, and give your comments/rating based on the rebuttal letter and the comments from other reviewers?
>
> Best Regards
>
> AC

---

### Decision · Program_Chairs · 2023-09-21

**Decision:**

Accept (poster)

**Comment:**

This paper receives comments from five reviewers. Though none of the reviewers rejects this paper, four of the reviewers gave borderline accept scores. The paper aims to bridge the gap between OOD detection and visual interpretation via using uncertainty of the model in explaining the own predictions. Reviewers pointed out that the method is simple and effective. Reviewers also recognized that the work provides a novel perspective on quantifying the disparities between ID and OOD data by analyzing the uncertainties in gradient-based attribution methods. Though the method is simple, it is interesting and effective. AC thus recommends accept at this stage, but urges the authors to improve the clarity and writing of the paper, as suggested by reviewers kfJE and CiZ8, and incorporate other suggestions to the final version.